



**Evaluating four gap-filling methods for eddy covariance measurements of**
**evapotranspiration over hilly crop fields**
Nissaf Boudhina [1, 2], Rim Zitouna-Chebbi [3], Insaf Mekki [3], Frédéric Jacob [1, 3], Nétij Ben
Mechlia [2], Moncef Masmoudi [2], Laurent Prévot [4]
[1] Institut de Recherche pour le Développement (IRD) – UMR LISAH (IRD, INRA,
Montpellier SupAgro), Montpellier, France
[2] Institut National Agronomique de Tunisie (INAT) / Carthage University, Tunis, Tunisia
[3] Institut National de Recherche en Génie Rural, Eaux et Forêts (INRGREF) / Carthage
University, Ariana, Tunisia
[4] Institut National de la Recherche Agronomique (INRA) - UMR LISAH (IRD, INRA,
Montpellier SupAgro), Montpellier, France
Corresponding author: Rim Zitouna-Chebbi, INRGREF – Carthage University, BP N°10,
Ariana 2080, Tunisia. (rimzitouna@gmail.com)



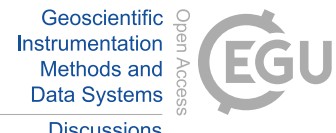

**Abstract**. Estimating evapotranspiration in hilly watersheds is paramount for managing water
resources, especially in semi-arid regions. Eddy covariance (EC) technique allows continuous
measurements of latent heat flux LE. However, time series of EC measurements often
experience large portions of missing data, because of instrumental dysfunctions or quality
filtering. Existing gap-filling methods are questionable over hilly crop fields, because of
changes in airflow inclination and subsequent aerodynamic properties. We evaluated the
performances of different gap-filling methods before and after tailoring to conditions of hilly
crop fields. The tailoring consisted of beforehand splitting the LE time series on the basis of
upslope and downslope winds. The experiment was setup within an agricultural hilly
watershed in northeastern Tunisia. EC measurements were collected throughout the growth
cycle of three wheat crops, two of them located in adjacent fields on opposite hillslopes, and
the third one located in a flat field. We considered four gap-filling methods: the REddyProc
method, the linear regression between LE and net radiation Rn, the multi-linear regression of
LE against the other energy fluxes, and the use of evaporative fraction EF. Regardless of
method, the splitting of the LE time series did not impact the gap filling rate, and it might
improve the accuracies on LE retrievals in some cases. Regardless of method, the obtained
accuracies on LE estimates after gap filling were close to instrumental accuracies, and were
comparable to those reported in previous studies over flat and mountainous terrains. Overall,
REddyProc was the most appropriate method, for both gap filling rate and retrieval accuracy.
Thus, it seems possible to conduct gap-filling for LE time series collected over hilly crop
fields, provided the LE time series are beforehand split on the basis of upslope / downslope
winds. Future works should address consecutive vegetation growth cycles for a larger panel of
conditions in terms of climate, vegetation and water status.
**Keywords**: Eddy covariance; latent heat flux; gap filling; hilly terrain; airflow inclination;
energy balance closure.



## 1. Introduction

Actual evapotranspiration is the amount of water transferred to the atmosphere by plant transpiration, soil evaporation, and vaporization of precipitation / condensation intercepted by plant canopies (Zhang et al., 2016). It directly drives biomass production, as photosynthesis is strongly linked to plant transpiration (Olioso et al., 2005). It is also a major term of land surface energy balance, since it is energetically equivalent to latent heat flux LE (Montes et al., 2014). Furthermore, it is a major term of water balance, since it represents up to 2/3 of the annual water balance for semi-arid and subhumid Mediterranean climates (Moussa et al., 2007; Yang et al., 2014). Therefore, determining actual evapotranspiration over land surfaces is important for managing agricultural activities.

Using evapotranspiration measurements for environmental and water sciences requires complete time series of latent heat flux LE at the hourly timescale, to be next converted into daily, monthly or annual values (Falge et al., 2001a; Falge et al., 2001b). This is a prerequisite for long-term studies in relation to global change, but also for short term studies in relation to agricultural issues and modeling challenges. However, common time series of eddy covariance (EC) measurements, which are nowadays considered as the reference method, include missing data because of experimental troubles such as power failures or instrumental dysfunctions. Also, unfavorable micro-meteorological conditions lead to reject significant parts of data that do not fulfill theoretical requirements for EC measurements. Statistical studies based on long-term measurements suggest that missing data rates range from 25 to 35% (Baldocchi et al., 2001; Falge et al., 2001a; Law et al., 2002), while data rejection rates through quality control range from 20% to 60% (Papale et al., 2006). Therefore, gap-filling methods are necessary to obtain continuous time series of land surface energy fluxes.

Most existing gap-filling methods were devoted to carbon dioxide ($CO_2$) measurements (Aubinet et al., 1999; Falge et al., 2001a; Goulden et al., 1996; Greco and





Baldocchi, 1996; Grünwald and Bernhofer, 1999; Moffat et al., 2007; Reichstein et al., 2005;
Ruppert et al., 2006). Table 1 summarizes the few studies that addressed measurements of
latent heat flux LE (Abudu et al., 2010; Alavi et al., 2006; Beringer et al., 2007; Chen et al.,
2012; Cleverly et al., 2002; Eamus et al., 2013; Falge et al., 2001b; Hui et al., 2004; Papale
and Valentini, 2003; Roupsard et al., 2006; Zitouna-Chebbi, 2009). The most usual gap-filling
methods are Look-Up Tables (LUT) based methods, Mean Diurnal Variation (MDV) method
and multivariate approaches. LUT based methods consist in filling gaps with data collected
under similar meteorological conditions. MDV based methods consist in replacing missing
values by the mean obtained on adjacent days. Multivariate approaches (i.e., artificial neural
networks, principle component analysis, interpolations and regressions) consist in filling gaps
using linear or non-linear relationships that involve drivers of evapotranspiration such as
meteorological variables, soil water content or net radiation. Prior to gap filling, time series
are often split in different ways according to the experimental conditions (e.g., nighttime /
daytime, wind directions, vegetation phenology, weekly or monthly time windows), so that
missing   data   are   filled   with   observations   collected   in   similar   conditions   for
micrometeorology, vegetation phenology and water status. Overall, gap-filling methods for
LE time series have been evaluated over flat, hilly and mountainous areas. However, the
existing studies for hilly areas did not address their specific conditions (Hui et al., 2004), or
they restricted the investigations to one gap-filling method only (Zitouna-Chebbi, 2009).

[Table 1 about here]

Hilly watersheds are widespread within coastal areas around the Mediterranean basin,

as well as in Eastern Africa, India and China. They experience agricultural intensification
since hilly topographies allow water-harvesting techniques that compensate for precipitation
shortage (Mekki et al., 2006). Their fragility is likely to increase with climate change and
human   pressure,   especially   as   water   scarcity   already   limits   crop   production.   Thus,



understanding evapotranspiration processes within hilly watersheds is paramount for the
design of decision support tools devoted to water resource management (McVicar et al.,

2007).

Gap-filling methods for LE have to be designed in accordance with the terrain

specificities that impact evapotranspiration. Conversely to flat terrains that correspond to
slope lower than 2% (Appels et al., 2016), solar and net radiations within sloping terrains
change depending on slope orientation, with larger values for ecliptic-facing slopes (Holst et
al., 2005). Over sloping terrains, the conditions of topography and airflow within the
atmospheric boundary layer (ABL) are very different for hilly areas as compared to
mountainous areas. Regarding topography, hilly areas depict lower slopes on average, and
Prima et al. (2006) proposed a threshold value of 22%. Regarding atmospheric stability, hilly
areas rise over small fractions of the daytime ABL, and the overlying airflows are slightly
influenced by stratification, which corresponds to neutral or instable conditions (Raupach and
Finnigan, 1997). Regarding wind regimes, externally driven winds are more frequent within
hilly areas, as compared to mountainous areas with anabatic and katabatic flows (Hammerle
et al., 2007; Hiller et al., 2008), and wind regimes differ much between the upwind and lee
sides of hills (Dupont et al., 2008; Raupach and Finnigan, 1997). Therefore, the relationships
on which rely the existing gap-filling methods, mostly co-variation of convective fluxes with
meteorological variables or temporal auto-correlation of the convective fluxes, are likely to
change with wind direction and vegetation development within hilly areas, because of
changes in airflow inclination (Zitouna-Chebbi et al., 2012; 2015), and therefore changes in
aerodynamic properties (Blyth, 1999; Rana et al., 2007).

In the context of obtaining continuous time series of evapotranspiration from EC

measurements of latent heat flux LE, the current study aimed to examine and compare LE
gap-filling methods over hilly crop fields. For this, we evaluated the performances of different



methods before and after tailoring to the conditions of hilly crop fields. We used the following
methodological framework.
• The experiment was set within a Tunisian agricultural hilly watershed with rainfed crops.
It included the data collection and preprocessing, the analysis of the experimental
conditions, and the analysis of the dataset to be filled.
• We considered several gap-filling methods that differ in the use of ancillary information,
either micrometeorological data or energy flux data other than LE. Given the possible
influence of airflow inclination, the gap-filling methods were tailored by splitting the
dataset on the basis of airflow inclination as driven by wind direction.
• We assessed the performances of the gap-filling methods by addressing (1) filling rate as
compared to missing data after preprocessing, (2) retrieval accuracy on filled data, and
(3) quality of gap-filled time series through energy balance closure.
**2. The experiment: study site and materials**
**2.1. Experimental site**
The Lebna watershed is located in the Cap Bon Peninsula, northeastern Tunisia. It extends
from the Jebel Abderrahmane to the Korba Laguna, and includes the Kamech watershed
(outlet at 36°52'30"N, 10°52'30"E, 108 m asl) that has an area of $2.7 \times 0.9$ km$^2$ (Figure 1). The
El Gameh wadi crosses Kamech from the northeast to the southwest. A hilly dam (140000 m$^3$
nominal capacity) is located at the watershed outlet. The Kamech watershed belongs to the
environmental research observatory OMERE (French acronym for Mediterranean
Observatory of Water and Rural Environment, http://www.umr-lisah.fr/omere).
[Figure 1 about here.]
The climate of the Kamech watershed is sub-humid Mediterranean. Over the [1995-
2014] period, yearly precipitation and Penman-Monteith reference crop evapotranspiration


(Allen et al., 1998) are 624 mm and 1526 mm, respectively. Terrain elevation ranges from
94 m asl to 194 m asl, and terrain slopes range between 0% and 30%, the quartiles being 6%,
11% and 18% (Zitouna-Chebbi et al., 2012). The soils have sandy-loam textures, and soil
depth ranges from few millimeters to two meters according to both the location within the
watershed and the local topography. These swelling soils exhibit shrinkage cracks under dry
conditions during the summer (Raclot and Albergel, 2006).

Within the Kamech watershed, agriculture is rainfed, traditional and extensive (Mekki

et al., 2006). Main crops are winter cereals (barley, oat, triticale, wheat), and legumes
(chickpeas, favabeans). Land use and parcels are strongly related to topography and soil
quality. The watershed includes 273 plots which sizes range from 0.08 to 13.65 ha (0.62 ha on
average, with a standard deviation of 1.05 ha).
**2.2. Measurement locations and experimental period**
Three flux stations simultaneously collected measurements of energy fluxes and
meteorological variables within three wheat crop fields (Figure 1): two sloping fields (A, B)
and a flat field (C).

Field A was located on the northern rim of the Kamech watershed. It had a fairly

homogeneous terrain slope (6°) that faced south-southeast, and a 1.2 ha area. Field B was
adjacent to field A, on the opposite hillside. It also had a homogeneous slope (5.2°) that faced
north, and had a 1 ha area. Fields A and B were separated by the northwestern limit of the
Kamech watershed. Field C was located in the southeastern part of the Kamech watershed. It
had a flat terrain and a 5 ha area. A meteorological station (labeled M in Figure 1) was located
near the watershed outlet.





The experimental period started at the beginning of December 2012 and ended mid-
June 2013. It thus covered the full growth cycle within the three wheat crop fields, from
sowing (1st December) to harvest (May 15 for field A, June 19 for fields B and C).
**2.3. Instrumental equipment and data acquisition**
On fields A, B and C, each flux station collected measurements of the land surface energy
fluxes (net radiation, soil heat flux, sensible and latent heat fluxes). Table 2 displays the type
of instruments used for each flux station along with acquisition and storage frequencies, and
sensor accuracies according to manufacturers.

[Table 2 about here.]

The sonic anemometers, krypton hygrometers, and air temperature and humidity
probes were installed at constant heights above ground level: 1.98 m for field A, 2.0 m for
field B and 2.2 m for field C. The verticality of the sonic anemometers was carefully checked
during the experiment with a spirit level, as described by Zitouna-Chebbi et al. (2012). The
latter reported a 1° accuracy on sonic anemometer verticality, according to the experimental
protocol and to the analysis of airflow inclination. To avoid water ponding on mirrors of the
krypton hygrometers, we rotated each of them in its mount so that the mirrors were vertical
and the measurement path was horizontal. The net radiometers were installed at 1.7 m height
above ground level and their horizontality was also checked regularly. For each flux station,
three soil heat flux sensors were distributed few meters around the station, and were buried at
5 cm below the soil surface.
Measurements at the meteorological station included: (1) solar irradiance with a
SP1100 pyranometer (Skye, UK); (2) air temperature and humidity with a HMP45C probe
(Vaisala, Finland); (3) wind speed with an A100R anemometer (Vector Instruments, UK); and
(4) wind direction with a W200P wind vane (Vector Instruments, UK). The instruments were


installed at 2 m above ground level (1 m for the pyranometer). All instruments were
connected to a CR10X data-logger (Campbell Scientific, USA) that calculated and stored the
30-minute averaged values from the 1 Hz frequency measurements.

All instruments were manufacturer-calibrated. Hereafter in the paper, we focused on

daytime measurements, since nighttime values of sensible and latent heat fluxes are small at
the daily timescale.

### 2.4. Data processing: calculation of net radiation and soil heat flux

On fields A and B, the measurements of net radiation (Rn) were corrected for the effect of
slope following the procedure proposed by Holst et al. (2005). Details are given in Zitouna-
Chebbi et al. (2012) and Zitouna-Chebbi et al. (2015). Only direct solar irradiance was
corrected by accounting for the angle between solar direction and the normal to local
topography. Direct solar irradiance was empirically derived from total solar irradiance
measured at the flux station. We characterized local topography with slope (topographical
zenith with nadir as origin) and aspect (topographical azimuth with north as origin), both
derived from a four-meter spatial resolution DEM obtained with a stereo pair of Ikonos
images (Raclot and Albergel, 2006). The correction for slope effect on Rn was about
50 W m$^{-2}$ on average.

For each flux station, soil heat flux (G) was estimated by averaging the measurements

collected with the three soil heat flux sensors. We did not apply any correction for heat
storage between the surface and the sensors for several reasons. First, the existing solutions
are questionable when considering swelling soils that exhibit shrinkage cracks under dry
conditions during the summer, since they require detailed and stable experimental protocols
(Leuning et al., 2012). Second, the experiment lasted throughout wheat growth cycles without
any flood event that are critical for heat storage correction. Third, neglecting the heat storage
in the soil above the heat flux plates induces errors on soil heat flux estimates that are not



systematically large, since they range between 20 and 50 W m$^{-2}$ on average (20-50% relative
to measured value), as reported by Foken (2008).

**2.5. Data processing: calculation of convective fluxes**

Sensible (H) and latent (LE) heat fluxes were calculated from the 20 Hz data collected by the
sonic anemometers and the krypton hygrometers, using the ECPACK library version 2.5.22
(Van Dijk et al., 2004). H and LE fluxes were calculated over 30 minute intervals.

**2.5.1. Flux calculation**

Most of the instrumental corrections proposed in the aforementioned version of the ECPACK
library were applied. These corrections addressed (1) the calibration drift of the krypton
hygrometer using air humidity and temperature measured by the HMP45C probe; (2) the
linear trends over the 30-min intervals; (3) the effect of humidity on sonic anemometer
measurement of temperature; (4) the hygrometer response for oxygen sensitivity; (5) the mean
vertical velocity (Webb term); (6) the corrections for path averaging and frequency response
(spectral loss); and (7) the rotation correction for airflow inclination (see Section 2.5.2).

**2.5.2. Coordinate rotations**

When calculating energy fluxes with the EC method, it is conventional to rotate the
coordinate system of the sonic anemometer (Kaimal and Finnigan, 1994). Coordinate
rotations were originally designed to correct the vertical alignment of the sonic anemometer
over flat terrains, and they are commonly used over non-flat terrains to virtually align the
sonic anemometer perpendicularly to the mean airflow, in an idealized homogeneous flow.
Common rotation methods are the double rotation and the planar fit method. In both methods,
the anemometer is virtually rotated around its vertical axis (yaw angle) to cancel the lateral
component of the horizontal wind speed.



The planar fit and double rotation methods calculate the rotations in different ways. In
the planar fit method (Wilczak et al., 2001), a mean streamline plane is evaluated by multi-
linear regression of the vertical wind speed (w) against the two horizontal components of the
wind speed (u and v). This multi-linear regression is applied over long periods, usually
several days or weeks. The double rotation method is applied to each time interval over which
the convective fluxes are calculated (30 minutes in our case). After the first rotation that
cancels the lateral component of the horizontal wind speed (yaw angle, see previous
paragraph), a second rotation (pitch angle) is applied around a horizontal axis perpendicular to
the main wind direction, to cancel the mean vertical wind speed. Thus, it implicitly accounts
for changes in wind direction and vegetation height that are likely to be constant over 30-
minute intervals.
Both double rotation and planar fit methods have advantages and drawbacks. On the
one hand, a significant variability in rotation angles can be observed at low wind speeds with
the double rotation method (Turnipseed et al., 2003). On the other hand, the planar fit method
must be applied for different sectors of wind direction and for different intervals of vegetation
height in case of sloping terrains and changes in vegetation height (Zitouna-Chebbi et al.,
2012; 2015). Since our study area was typified by large wind speeds (Zitouna-Chebbi et al.,
2012; 2015), we selected the double rotation method.
**2.5.3. Data quality assessment**
Several quality criteria for flux measurements have been proposed in the literature. The most
commonly used are the steady-state (ST) test and the integral turbulence characteristics (ITC)
test (Foken and Wichura, 1996; Geissbühler et al., 2000; Hammerle et al., 2007; Rebmann et
al., 2005). These tests verify that the theoretical requirements for the EC measurements are
fulfilled. The ST test assesses the homogeneity of turbulence over time, while the ITC test
assesses the spatial homogeneity of turbulence. Although established over flat terrains, they




have been used for long over mountainous terrains (Hammerle et al., 2007; Hiller et al., 2008)
and more recently over hilly terrains (Zitouna-Chebbi et al., 2012; 2015), because there is no
specific test for relief conditions.

Quality classes were assigned to each half-hourly flux data according to the results of

the two tests. For this, we followed the classification proposed by Foken et al. (2005) and
Rebmann et al. (2005). H and LE flux data belonging to the quality class I could be used for
turbulence studies. H and LE flux data belonging to classes II to IV could be used for long-
term flux measurements. Finally, we rejected H and LE flux data belonging to class V that
correspond to both ST > 0.75 and ITC > 2.5.

Regarding footprint, the flux contributions were likely to originate from the target

fields, regardless of wind direction and vegetation height. On the one hand, experimental
conditions (measurement height, field size, vegetation height and micrometeorology) were
similar to those indicated in Zitouna-Chebbi et al. (2012) and Zitouna-Chebbi et al. (2015).
On the other hand, the latter reported that calculated flux contribution from the target fields
were about 75%-80% throughout three one-year duration experiments. In the next section, we
address the vegetation and micrometeorological conditions, as well as the subsequent
relevance of measurement height.
**2.6. Experimental conditions**
**2.6.1. Climate forcing and wind regime**
During the experiment that lasted from December 2012 to June 2013, the meteorological
station (M) recorded a cumulative precipitation of 563 mm. Over the same period, the
reference evapotranspiration $ET_0$ recorded by the meteorological station ranged between
1.1 and 5.8 mm day$^{-1}$ at the daily timescale, with a cumulated value of 510 mm.





The wind speed value recorded during the experimental period by the meteorological
station was 4 m s$^{-1}$ on average. This value was as twice as the worldwide value over lands
(Allen et al., 1998). The averaged wind speed value recorded by the meteorological station
was very close to those recorded by the sonic anemometers installed on the flux stations
within field A, B and C, with differences lower than 0.4 m s$^{-1}$. The spatial homogeneity for
wind speed was also observed in previous studies conducted on different locations within the
same watershed (Zitouna-Chebbi, 2009; Zitouna-Chebbi et al., 2012; 2015).
The wind rose obtained from the data collected at the meteorological station depicted
two prevailing directions (Figure 2). The first direction corresponded to winds coming from
south (directions between 70° and 220°, clockwise, North is 0°). The second direction
corresponded to winds coming from the other directions, hereafter referred to as northwest
winds. The topography induced downslope winds on field A and upslope winds on field B
under northwest winds. The reverse was observed under south winds.
[Figure 2 about here.]
Micrometeorological conditions were analyzed using the atmospheric stability
parameter $\xi = (z-D) / L_{MO}$, where z is measurement height, D is displacement height and $L_{MO}$
is Monin-Obukhov length. D was set as two third of vegetation height, the latter being derived
from in-situ measurements (see Section 2.6.2). The atmospheric stability parameter $\xi$ was
most of the time negative, with notably few values larger than 0.1, mainly during sunrise or
sunset. The $\xi$ median values were −0.007, −0.011 and −0.010 respectively for field A, B and
C. These values corresponded to conditions of forced convection (neutrality or low instability)
induced by large wind speeds. We did not observe notable differences between northwest and
south winds. Zitouna-Chebbi et al. (2012) and Zitouna-Chebbi et al. (2015) obtained similar
results with a dataset collected between 2003 and 2006 on different fields within the same
study area.
Overall, the analysis of wind direction and micrometeorological conditions indicated
that the wind regime did not stem from valley wind or sea breeze. Indeed, the wind direction
did not depict any diurnal course in relation to anabatic / katabatic flows or to sea / land heat
transfers, while the ξ parameter did not correspond to conditions of atmospheric stability with
free convection.
**2.6.2. Vegetation conditions**
Throughout the experiment, the evolution of the wheat phenology was monitored using the
scale of Feekes and Large so-called "BBCH Scale improved" (Lancashire et al., 1991). Fields
A, B and C depicted similar phenological evolutions. The beginning of tillering stage
appeared on January 15, and full tillering was on February 19. Start of bolting was on
March 5, and full flowering was on April 22. Seed maturity stage lasted from the beginning to
the end of May, and the beginning of senescence was late May.
Vegetation height was measured on a weekly basis using a tape measure. For each
date, 30 height measurements were performed within each field, and next averaged at the field
scale. Vegetation height reached its maximum on April 22, and maximum averaged values
were 1.00 m, 0.87 m and 0.98 m, for fields A, B and C, respectively. Vegetation height
measurements were next interpolated on a daily basis by using a logistic function.
The vegetation height data indicated that the sonic anemometers and KH20 krypton
hygrometers, set up around 2 m above soil surface, was located above the roughness sublayer.
Indeed, the experiment was typified by neutral or slightly unstable conditions that
corresponded to a roughness sublayer extension from the ground up to 1.43 × vegetation
height (Pattey et al., 2006).
Green leaf area index (LAI) was measured using a planimeter. Every two weeks, all
leaves were collected within three one-meter-long transects to derive a spatially averaged



value. LAI reached its maximum on April 11, and maximum values were 2.5 m²/m²,
2.3 m²/m² and 2.3 m²/m² for fields A, B and C respectively.

**2.7. The dataset to be filled**

Missing LE data stemmed from (1) total shutdowns of flux stations, following battery
discharges or vandalism acts; (2) dysfunctions of KH20 krypton hygrometers after
precipitation events when air humidity permeated the sensor because of seal degradation; and
(3) rejection of LE data identified as class V data by ST and ITC tests (Section 2.5.3).

Table 3 displays the amounts of available data derived from EC measurements over

the three fields, when considering the latent heat flux (LE). It gives the beginning and ending
dates of the EC measurements, the number of daytime data over 30 minutes intervals, the
numbers and proportions of data with good (classes I to IV) and bad quality (class V)
according to ST and ITC tests, the number of missing data due to dysfunctions of the Krypton
hygrometer (KH20), and the number of missing data because of total shutdown of flux
stations.

[Table 3 about here.]

The ratio of acquired LE data after filtering ranged between 20 % and 61 %. It was

rather low as compared to the ratios reported by former studies at the yearly timescale for
worldwide flux networks such as FLUXNET (65%), where these ratios stemmed from system
failures or data rejection (Baldocchi et al., 2000; Falge et al., 2001a; Falge et al., 2001b). The
low ratio we obtained in the current study was ascribed to KH20 dysfunctions and total
shutdown of flux stations. Furthermore, the KH20 sensor installed on field B was out of order
from the end of March until the end of the experiment, because of severe instrumental
dysfunctions.



The proportion of bad quality data was low, with around 3 % of data belonging to
class V. The results of the quality control tests did not exhibit any difference between the
fields. For sensible heat flux H, the percentages of data belonging to the high quality classes
(I to IV) were 85 %, 84 % and 88 % for fields A, B and C, respectively.
On the one hand, the rate of missing data for the current study, between 40 and 80%,
was much larger than those reported in former studies, i.e., between 25 and 35% (Baldocchi et
al., 2001; Falge et al., 2001a; Law et al., 2002). On the other hand, the rate of rejected data by
quality control, between 2 and 4%, was much lower for the current study as compared to
those reported in former studies, i.e., between 20 and 60% (Papale et al., 2006). Therefore, the
overall rate of data to be filled was comparable to those reported in former studies.
**3. Methods**
**3.1. Rationale in choosing and implementing gap-filling methods**
Amongst the existing LE gap-filling methods listed in Introduction (Table 1), we selected
some methods that differ in the use of ancillary information, either meteorological variables
or energy fluxes. The meteorological data to be used were those provided by the
meteorological station, while the flux data to be used were those collected at each of the three
flux stations of interest (Section 2.3). We did not select methods that involve measurements of
soil water content or vegetation canopy, since energy fluxes indirectly account for the latter at
a spatial scale closer to that of the LE missing data (see results about footprint analysis in last
paragraph of Section 2.5.3).
Amongst the existing LE gap-filling methods listed in Introduction (Table 1), we
selected the commonly used REddyProc method that relies on LUT and MDV to fill missing
flux data with those collected under similar meteorological conditions or with averaged values
over adjacent days. We also selected methods that fill LE gaps by using multilinear



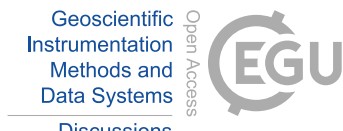

regressions on other energy flux data (Rn, H and G). We did not select methods based on
artificial neural networks because ensuring the relevance of calibration, testing and validation
steps require large datasets of at least one year (Abudu et al., 2010; Beringer et al., 2007;
Eamus et al., 2013; Papale and Valentini, 2003).

**3.1.1. REddyProc**

For the REddyProc method, we selected the online tool available at http://www.bgc-
jena.mpg.de/REddyProc/brew/REddyProc.rhtml, and that is based on Reichstein et al. (2005).
The REddyProc method combines the co-variation of the convective fluxes with
meteorological variables (Falge et al., 2001b) and the temporal auto-correlation of the
convective fluxes (Reichstein et al., 2005). Gaps are filled in accordance with available
information by considering three cases: (1) solar radiation (Rg), air temperature (Tair), and
vapor pressure deficit (VPD) data are available; (2) Rg data only are available; and (3) none
of the Rg, Tair, VPD data are available.
• For Case (1), the missing LE value is replaced by the average value under similar
meteorological conditions within a time window of ±7 days. Similar meteorological
conditions correspond to Rg, Tair and VPD values that do not deviate by more than
50 W m$^{-2}$, 2.5 °C, and 5 hPa, respectively. If no similar meteorological conditions occur
within the ±7 day time window, the latter is extended to ±14 days.
• For Case (2), a similar approach is taken. Similar meteorological conditions correspond to
Rg deviation by less than 50 W m$^{-2}$, and the window size is not extended.
• For Case (3), the missing value is replaced by an adjacent value within ±1 hour, or by an
averaged value at the same time of the day that is derived from the mean diurnal course
over ±1 day.
In case the three steps do not permit to fill the gaps, the whole procedure is repeated while
increasing the window sizes until the value can be filled. Thus, the window size increases





using 7-day steps until ±70 days for Case 1 and 2, and until ±140 days for Case 3, which
obviously result in a degradation of the quality indicator.

### 3.1.2. LE reconstructed from Rn

Initially proposed by Cleverly et al. (2002), this method was successfully tested on our study
site by Zitouna-Chebbi (2009). It assumes the stability of the LE / Rn ratio over a given
period that can be one day, one month or one year (Table 1). We implemented the method by
first calibrating the linear regression LE = a Rn + b on existing LE and Rn data, and next
applying the regression to missing LE data for which Rn was actually measured. This method
will be referred to as 'LE - Rn method' hereafter.

### 3.1.3. LE reconstructed from multi-linear regression against other energy fluxes

This method is an extension of the LE - Rn method, since LE is estimated as a linear
combination of the other energy fluxes Rn, H and G. As for the LE - Rn method, the multi-
linear regression (MLR) method was implemented by first calibrating the multi-linear
regression on existing LE, Rn, H and G data (LE = a' Rn + b' G + c' H + d'), and next
applying the regression to missing LE data for which the three other fluxes were actually
measured. This method will be referred to as 'MLR method' hereafter.
Energy balance theoretically implies a' = 1, b' = -1, c' = -1 and d' = 0. However, this
is not the case in practice because of the "energy imbalance problem" for EC measurements.
This problem has been mentioned in the literature for vegetated canopies and bare soils, as
well as over flat, mountainous and hilly terrains (Foken, 2008; Hammerle et al., 2007;
Leuning et al., 2012; Wilson et al., 2002; Zitouna-Chebbi et al., 2012; 2015). As reported by
Leuning et al. (2012), the energy imbalance problem is that the sum of the convective flux
(H + LE) underestimates available energy (Rn - G), because of theoretical assumptions (e.g.,
neglecting storage terms or lateral turbulent transfers) and because of experimental





assumptions (e.g., neglecting measurement inaccuracies, neglecting differences in
measurement spatial extensions). Thus, applying the energy balance equation LE = Rn - G - H
would transfer energy imbalance onto LE estimates, which is not the case with the MLR
method that involves a regression calibration (a' ≠ 1, b' ≠ -1, c' ≠ -1 and d' ≠ 0).

### 435     3.1.4. LE reconstructed from evaporative fraction (EF)

Evaporative fraction EF is defined as the ratio of latent heat flux LE over available energy
(Rn - G) when assuming the latter equals the sum of convective fluxes (H + LE). Li et al.
(2008) and Shuttleworth et al. (1989) showed that EF was almost constant during daytime
hours. Although rebutted (Hoedjes et al., 2008; Van Niel et al., 2011), various studies stated
that EF at midday ($EF_{md}$) is statistically representative of daily EF, and thus recommended to
use $EF_{md}$ for estimating LE (Crago and Brutsaert, 1996; Crago, 1996; Gentine et al., 2011; Li
et al., 2008; Peng et al., 2013).

The estimation of missing LE data was twofold. In a first step, $EF_{md}$ was calculated on

a daily basis by using the measured data over the four hours centered on solar noon, provided
that 75% at least of the eight 30 minutes data was available between noon -2h and noon +2h
for LE, Rn, and G.

$$EF_{md} = \sum_{noon-2h}^{noon+2h} LE_i \bigg/ \sum_{noon-2h}^{noon+2h} (Rn_i - G_i)$$

In a second step, the missing LE data were estimated as LE = (Rn - G) $EF_{md}$, when Rn and G
were actually measured. This method will be referred to as 'EF method' hereafter.

As compared to the MLR method that implicitly accounts for the energy imbalance

problem via the regression calibration, the EF method induced an overestimation of the
convective fluxes, by replacing H + LE with available energy Rn - G. Conversely, averaging
EF around solar noon rather than over the diurnal cycle might induce an underestimation of

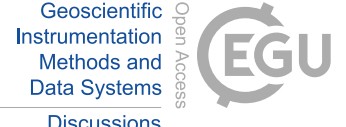

EF at the daily timescale. Therefore, the EF method was likely to (1) induce some errors on
LE estimates used for filling gaps, and (2) increase energy imbalance for the reconstructed
data because of the difference between H + LE and Rn - G.

**3.2. Tailoring the gap-filling methods to the conditions of hilly crop fields**

The gap-filling methods were tailored to the conditions of hilly crop fields by splitting the
dataset on the basis of the airflow inclination that is driven by the combined effect of wind
direction, topography and vegetation height. The analysis of the experimental conditions
showed that the wind regimes was typified by two main wind directions, i.e. northwest and
south, that induces upslope and downslope winds on field A and B (Section 2.6.1). Therefore,
any of the three datasets for field A, B and C was split into two sub-datasets that correspond
to northwest and south winds. We recall that (1) northwest winds correspond to downslope
and upslope winds on field A and B, respectively, (2) south winds correspond to upslope and
downslope winds on field A and B, respectively, and (3) field C was horizontal.
Most existing gap filling methods for LE measurements include a prior splitting of the
time series to be filled (Table 1), so that missing data are filled with existing observations
collected under similar conditions (e.g., nighttime / daytime, wind directions, vegetation
phenology, weekly or monthly time windows). REddyProc relies on time windows ranging
from 1 to 140 days with Case 1 and 2, and up to 280 days with Case 3 (Section 3.1.1). The EF
method relies on an estimate of evaporative fraction for each day, and therefore implicitly
splits the time series on a daily basis. The LE - Rn method assumes that the linear relation
between LE and Rn is stable over time, and the MLR method assumes that the multi-linear
regression between LE, Rn, G and H is also stable over time. For both LE - Rn and MLR
methods, it was therefore necessary to split the time series into nominal periods over which
the regressions were likely to be stable. This was all the more necessary since vegetation



development can combine with wind direction and thus impact the regression between LE and
other energy fluxes.

For both the LE - Rn and MLR methods, we split the dataset into three periods that

differed in vegetation phenology. By splitting the dataset on the basis of vegetation
phenology, we indirectly accounted for changes in soil water content and vegetation height at
monthly to seasonal timescales. The beginning and ending of each period are given in
Table 4, along with the vegetation and climatic conditions. The first period corresponded to
active green vegetation, with moderate reference evapotranspiration, and with abundant and
frequent precipitation events that supply plant transpiration and soil evaporation. It was
typified by the absence of water stress, and therefore large values for both evaporative
fraction EF and LE / Rn ratio. We labeled this first period "GV" for green vegetation. The
second period preceded grain maturation and leaf senescence. It corresponded to the
beginning of water stress that resulted from the combined effect of limited precipitation and
large reference evapotranspiration. We labeled this second period "PS" for pre-senescence.
The third period corresponded to leaf senescence and grain maturation. It corresponded to a
pronounced water stress that resulted from the combined effect of no precipitation and large
reference evapotranspiration. We labeled this third period "SV" for senescent vegetation.

[Table 4 about here.]

**3.3. Assessing the performances of the gap-filling methods**
The performances of the three gap-filling methods were assessed on filling rate, retrieval
accuracy and quality of gap-filled time series through energy balance closure. In order to
make comparable the performances of the four methods, we used the following procedure.



- Conversely to REddyProc, the LE - Rn, MLR and EF methods were not able to fill gaps induced by total shutdowns of the flux stations. Therefore, we addressed the filling of the gaps that resulted from dysfunctions of the KH20 sensors and quality filtering only.

- For field B, the LE - Rn, MLR and EF methods were not able to fill gaps induced by the shutdown of the KH20 sensor from the end of March (middle of the GV period) to the end of experiment. Indeed, the EF method required Rn, G and LE data on a daily basis, while the LE - Rn and MLR methods required data for each of the periods GV, PS and SV, which excluded periods PS and SV. Therefore, we disregarded the time period in question (from the end of March to the end of experiment) for field B.

- The filling performances were given in accordance with the number of reconstructible data (LE missing data because of both KH20 dysfunctions and quality filtering). They were expressed as the ratio of reconstructed to reconstructible data.

- The prior splitting of the time series to be filled is a common procedure for most gap-filling methods (Table 1), but is different from one method to another (Section 3.2). Therefore, we did not assess the performances of the gap-filling methods on the basis of the time periods GV, PS and SV. We discriminated the periods GV, PS and SV for the regression calibrations only (LE - Rn and MLR methods).

- To quantify retrieval accuracy, REddyProc provides estimates for each existing data, where the estimate is derived independently of the corresponding data. Therefore, we implemented a leave-one-out cross-validation (LOOCV) procedure to evaluate the retrieval accuracy for the LE - Rn, MLR and EF methods. For this, any estimate for retrieval accuracy was calculated by removing the corresponding reference value.

- We evaluated the performances of the gap filling methods before and after the splitting of the time series on the basis of wind direction (northwest / south). We separately



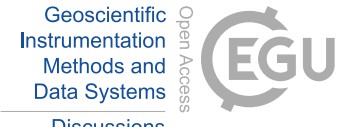
considered the field A, B and C, where field A and B are located on two opposite hillsides
with upslope and downslope winds, and field C is located on a horizontal terrain.
•   The retrieval accuracy was quantified using absolute and relative root mean square error
(RMSE and RRMSE) as well as mean absolute difference (MAE), bias and coefficient of
determination $R^2$ (Jacob et al., 2002; Moffat et al., 2007).
•   To evaluate the quality of the gap-filled time series, we compared the sum of the
convective energy fluxes (H + LE) against available energy (Rn - G) before and after gap
filling, where gap filling was conducted after the splitting of the time series on the basis of
wind direction. Although energy balance closure analysis is questionable for assessing the
consistency of flux measurements, it permits to compare independent measurements.
**4. Results**
**4.1. Filling performances of the gap-filling methods**
For the three fields (A, B, C) and the two wind directions (northwest, south), Table 5 displays
the number of reconstructible data (LE missing data because of KH20 dysfunctions or LE
data belonging to quality class V), as well as the number and percentage of reconstructed data
by the four methods (REddyProc, LE - Rn, MLR and EF). For each field, the total number of
reconstructible data is also indicated, as well as the total number and corresponding
percentage of reconstructed data. The total number of reconstructible data in Table 5
corresponds to that given in Table 3 (i.e. sum of LE missing data because of KH20
dysfunctions and of LE data belonging to quality class V), apart from field B (2083 versus
3060) for which we restricted the time period to the GV period, since no LE data were
available on periods PS and SV because of the KH20 shutdown (second item in Section 3.3).

[Table 5 about here.]





With both the REddyProc and LE - Rn methods, all the missing LE data could be

reconstructed. The MLR method permitted to reconstruct 84%, 86% and 90% of the missing
LE data, on fields A, B and C respectively. The EF method permitted to reconstruct 32%,
19% and 70% of the missing LE data, on fields A, B and C respectively. The reconstruction
rates obtained with the MLR method were similar on fields A, B and C. On the other hand,
the reconstruction rate with the EF method was much larger on field C (flat terrain) than those
on field A and B (sloping terrains). Overall, the filling rate was the same for a given field,
whether we split or not the time series on the basis of wind direction.
**4.2. Accuracy of the gap-filling methods**
The calibration of the LE - Rn method for the three periods (GV, PS and SV) was similar for
fields A (Figure 3), B and C (Figure SP1a and SP1b in supplementary materials). The LE / Rn
ratio exhibited a notable temporal stability for each of the three periods, and we did not
observe any distinct scatterplot for the period GV, even if the scattering was larger as
compared to the periods PS and SV. On the other hand, we observed significant differences in
slope and offset from one period to another, with changes in slope between 90 and 170%
(relative to mean value), and changes in offset between 60 and 120% (relative to mean value).

[Figure 3 about here.]

We obtained similar LE - Rn regressions for field A (Figure 3) and B (Figure SP1a in

supplementary materials) when splitting the time series on the basis of south and northwest
winds that correspond to upslope (respectively downslope) and downslope (respectively
upslope) winds on field A (respectively B). Apart from the SV period with too few data on
field A, we noted some differences in regressions between the two wind directions for any
period, with changes in slope between 5 and 50% (relative to mean value), and changes in
offset between 40 and 80% (relative to mean value). On the other hand, the differences were
lower on field C with a flat terrain (see Figure SP1b in supplementary materials), with



changes in slope between 0.5 and 10% (relative to mean value), and changes in offset between
15 and 30% (relative to mean value). A covariance analysis conducted on the regression
coefficients showed that the changes in slope and offset were statistically significant in most
cases (Table SP1 in supplementary materials).

We quantified the retrieval accuracies of the four gap-filling methods by comparing

reference data and gap-filling retrievals of latent heat flux LE over 30 minute intervals for
each field and each wind direction (Table 6). The retrieval accuracies were obtained using a
LOOCV procedure (Section 3.3). We observed the following trends.
• The four methods provided similar retrieval accuracies, with differences between RMSE

values lower than 20 W m$^{-2}$. Bias values were almost null, apart from the EF method. In a

lesser extent, the RMSE values were lower with REddyProc that also provided better R²

values, and the EF method provided the larger RMSE and biases values, down to -

20 W m$^{-2}$ for bias.

• Regardless of gap-filling method, the retrieval accuracies were similar for field A and C,

whereas they were lower for field B.

• The method performances could be either different or similar before and after the splitting

of the time series on the basis of wind direction. For field A, the RMSE values were

similar for upslope and downslope winds, and they were comparable to those obtained

before the splitting. For field B, the RMSE values were much lower (respectively slightly

larger) for downslope winds (respectively upslope winds) as compared to those obtained

before splitting the time series. For field C with a flat terrain, the statistical indicators were

comparable before and after the splitting.

[Table 6 about here.]

**4.3. Energy balance closure analysis**

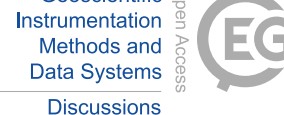

We recall that the gap-filling retrievals we considered for energy balance closure analysis
were those obtained with the splitting the time series on the basis of wind direction
(Section 3.3). We obtained similar results for energy balance closure for field A (Figure 4),
field B and C (Figure SP2a and SP2b in supplementary materials). Before and after gap
filling, the sum of the convective fluxes systematically underestimated available energy, apart
from field B after gap filling with the EF method. On a field basis, change in energy balance
closure from one gap-filling method to another was 15% for field A, 65% for field B, and
44% for field C, according to changes in the H + LE versus Rn - G regression slope. On a
method basis, energy balance closure varied from 5% (MLR) to 32% (EF) from one field to
another, according to changes in the H + LE versus Rn - G regression slope. Finally, energy
balance closure could be better after gap filling, and energy balance closure on sloping fields
A and B was comparable to that on the flat field C.

[Figure 4 about here.]

When comparing energy balance closure after gap filling with the four methods, we

could not identify any clear trend on the basis of the (H + LE) versus (Rn - G) linear
regression. Gap filling with the LE - Rn method provided among the best energy balance
closure, and gap filling with the REddyProc method provided among the worst energy
balance closure. Energy balance closure was very similar for the LE - Rn and MLR methods,
with changes in the regression slope between 2.5% (field C) and 4.5% (field A). Further, the
EF method could provide the worst (Field A) or the best (Field B) energy balance closure.
The scattering around the (H + LE) versus (Rn - G) regression was reduced after gap filling,
either slightly with the REddyProc and EF methods, or much with the LE - Rn and MLR
methods.
**5. Discussion**
**5.1. Filling performances of the gap-filling methods**





The filling rate was maximal with the REddyProc and LE - Rn methods. Indeed, REddyProc
relied on existing LE values within a given time window, either corresponding to similar
meteorological variables or derived from averaged diurnal courses. Similarly, the LE - Rn
method relied on continuous measurements of net radiation. The MLR method was less
efficient than the REddyProc and LE - Rn methods, because of both missing H measurements
and H data rejection by quality control. In this case, the filling rate was comparable to the
percentage of available H data given in Section 2.7 (84%, 86% and 90% versus 85%, 84%
and 88% for field A, B and C, respectively). The worst efficiency of the EF based gap-filling
method was explained by the fact that Rn, G and LE data around solar noon are required on a
daily basis.

The filling rate was similar whether we split or not the time series on the basis of wind

direction. For REddyProc, this was explained by the capability of the method to find LE data
under similar meteorological conditions or to obtain averaged values from diurnal courses
within a scalable time window. For the LE - Rn and the MLR methods, this was explained by
existing data for regressions within the three periods GV, PS and SV, when applicable. For
the EF method, this was explained by the daily basis computation of EF and the subsequent
filling at the daily timescale. Overall, the four methods were able to complete time series, in
spite of larger gap occurrences induced by the splitting of the time series on the basis of wind
direction. Also, it is important to note that conversely to the LE - Rn, MLR and EF methods
that relied on energy fluxes (Rn, G and H), REddyProc had the capability to fill gaps induced
by total shutdowns of the flux stations, although we did not address these total shutdowns to
make comparable the performances of the four methods.

We could not compare the filling rates we obtained in the current study against

outcomes from the former studies listed in Table 1 for LE data, owing to the absence of
information on this issue. The same applied for former studies about carbon dioxide.




### 5.2. Accuracy of the gap-filling methods

When calibrating the LE - Rn method, it was relevant to split the time series into the three periods GV, PS and SV, because of large changes in the LE - Rn regression from one period to another. The strong decrease of LE / Rn ratio throughout period GV to SV was ascribed to the decrease in LE magnitude because of vegetation senescence that combined with no precipitation and increasing reference evapotranspiration. This emphasized the impact of changes in soil water content and vegetation canopy at monthly to seasonal timescales. When calibrating the LE - Rn method, it was also relevant to split the time series on the basis of northwest and south winds. Indeed, some differences were observed between the two wind directions for the periods GV and PS, and these differences were larger for sloping terrains (fields A and B) than for the flat terrain (field C). As compared to former studies listed in Table 1, these outcomes were consistent with those from Zitouna-Chebbi (2009). Indeed, the latter reported the need to split time series into distinct periods and wind directions, so that it was possible to take into account changes in aerodynamic conditions for measurements collected within the same study area, over other crop fields and during other years.

The slightly better accuracies obtained with REddyProc indicated that this method was able to find appropriate LE values under similar meteorological conditions or within a given time window, in spite of possible changes in soil water content. LE - Rn and MLR provide very similar accuracies. We expected that MLR would outperform LE - Rn because of the additional inclusion into the regression of G and H fluxes that are driven by vegetation canopy and soil water content. Then, the similar accuracies might result from too large time windows for periods GV, PS and SV, and especially for period GV with large scattering around the regression line (see for instance Figure 3 with the LE - Rn regression). The EF method provided the lower accuracies. We expected better accuracies with the EF method that filled gaps on a daily basis, and the underperformance might result from the combination of (1) the


EF underestimation at the daily timescale when computed between 10:00 and 14:00 solar
time, and (2) the overestimation of H + LE by Rn - G as a result of energy imbalance. Overall,
the method performances were driven by the temporal dynamics of the local conditions in
terms of micrometeorology, vegetation canopy and soil water content. For instance, large
precipitations were likely to induce sharp changes in soil water content, thus advantaging the
EF method that is based on a daily basis computation, and disadvantaging the REddyProc
method that relies on similar meteorological conditions or average diurnal courses.

Overall, the RMSE values between reference data and gap-filling retrievals of latent

heat flux LE ranged between 20 W m$^{-2}$ and 90 W m$^{-2}$, and almost 2/3 of these values were
lower than 50 W m$^{-2}$. The retrieval accuracy was similar for the four gap-filling methods, and
was comparable to those reported by the previous studies listed in Table 1 (e.g. between 25
and 50 W m$^{-2}$ for RMSE).

Finally, the performances could be better when splitting the time series on the basis of

northwest and south winds, with much lower RMSE values for downslope winds. This was
not systematic for the sloping fields, but it was systematic for all methods when applicable,
although these methods involved different information for the reconstruction of the missing
data. Thus, our study confirmed that it may be relevant to discriminate upslope and
downslope winds when implementing gap-filling methods. This is consistent with reports
from Zitouna-Chebbi et al. (2012) and Zitouna-Chebbi et al. (2015) who showed the need to
discriminate upslope and downslope winds when correcting the influence of airflow
inclination on measurements collected over hilly crop fields.
**5.3. Energy balance closure analysis**
For the LE - Rn and MLR methods, energy balance closures were similar, they varied little
from one field to another, and they were better than those obtained with REddyProc and EF
methods. This was ascribed to the constraint on energy balance closure when replacing gaps



with LE estimates derived from regression between energy balance fluxes (LE versus Rn on
the one hand, and LE versus Rn, G and H on the other hand). Energy balance closure was
lower with REddyProc, and varied much from one field to another. This was ascribed to the
lack of constraint on energy balance closure when replacing gaps with LE data collected at
different times. For EF, energy balance closure varied much from one field to another, and
especially on field B with (H + LE) overestimating (Rn - G). This might be explained by
changes in compensation effects between (1) the EF underestimation at the daily timescale
when computed between 10:00 and 14:00 solar time, and (2) the overestimation of H + LE by
Rn - G as a result of energy imbalance.

For the four gap-filling methods, energy balance closure after reconstruction of the LE

data was comparable to that observed before gap filling, which showed the consistency of the
gap-filled time series. Further, energy balance closure for the two sloping fields (A and B)
was comparable to that obtained on the flat field (C), which showed the consistency of the
reconstructed data after the splitting of the time series on the basis of upslope / downslope
winds. We could not compare the energy balance closures we obtained in the current study
against the outcomes from to the former studies listed in Table 1 for LE data, owing to the
absence of information on this issue. Nevertheless, our values of energy balance disclosure
([15% - 35%]) were comparable to those reported in the literature ([10% - 30%]) for flat, hilly
and mountainous terrains (Foken, 2008; Hammerle et al., 2007; Li et al., 2008; Wilson et al.,
2002; Zitouna-Chebbi et al., 2012; 2015).
**6. Conclusion**
For the four gap-filling methods we evaluated (REddyProc, LE - Rn, MLR and EF), the
retrieval accuracies were similar and comparable to instrumental accuracies. On the other
hand, the filling rate was maximal for REddyProc and LE - Rn, whereas it was lower for
MLR and EF. Therefore, the REddyProc and LE - Rn methods were the most appropriate for



our study case, in terms of completing time series as much as possible while providing
retrievals with good quality. This outcome applied even more for the REddyProc method that
is able to fill gaps induced by total shutdowns, although a deeper analysis is beforehand need
to evaluate the retrieval accuracies in such situations.

Our results led us to recommend the splitting of LE time series on the basis of wind

direction, prior to the implementation of the gap-filling methods. Indeed, the prior splitting of
time series on the basis of wind direction might improve retrieval accuracies, although the
benefit was not systematic. Besides, the obtained accuracies on LE estimates after gap filling
were comparable to those reported in the literature for flat and mountainous areas, and the
same applied for energy balance closure as a consistency indicator for the filled time series.
Finally, the splitting of the time series did not impact the gap filling rate, in spite of larger gap
occurrences. Therefore, we conclude that it possible to conduct gap filling for time series
collected over hilly terrains, provided the prior splitting of the time series is applied in an
appropriate manner by discriminating upslope and downslope winds.

Our study case is widespread within the Mediterranean basin, because of orography

and climate conditions within coastal areas across the Mediterranean shores. In a lesser extent,
the outcomes of our studies are also of potential interest for hilly watersheds in Eastern
Africa, India and China. On the other hand, the experiment on which relied the current study
lasted over one crop growth cycle only, and we offset this temporal restriction by
simultaneously considering three locations that differed much in topographical conditions and
resulting airflow inclination. Nevertheless, future works should strengthen the outcomes of
the current study, by addressing (1) a larger panel of environmental conditions in relation to
climate, vegetation type and water statuses, and (2) consecutive vegetation growth cycles.
**Acknowledgments**



This study was supported by the IRD JEAI JASMIN-Tunisia project (INRGREF, INAT, and
LISAH), the IRD / ARTS program, the MISTRALS / SICMED project, the Agropolis
Foundation (contract 0901-013), and the ANR TRANSMED ALMIRA project (contract
ANR-12-TMED-0003-01). The ORE OMERE is thanked for providing the meteorological
data. We are extremely grateful to Dr. Tim McVicar for constructive discussions that helped
to improve the manuscript.

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





**List of Figures**



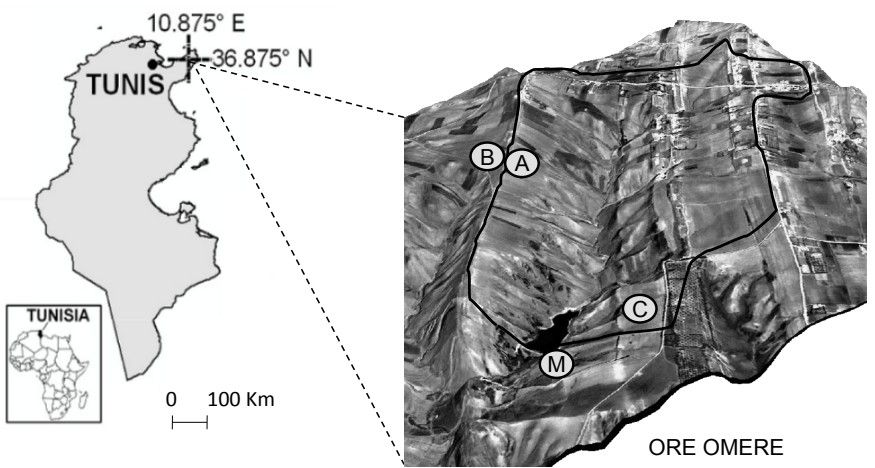

Figure 1. Location of the Kamech watershed within the Cap Bon Peninsula, north eastern
Tunisia (left). Kamech has a 0.9 km width and a 2.7 km length. Three-dimensional view of
Kamech (right), including locations of the experimental fields (A, B, C) and of the standard
meteorological station (M).





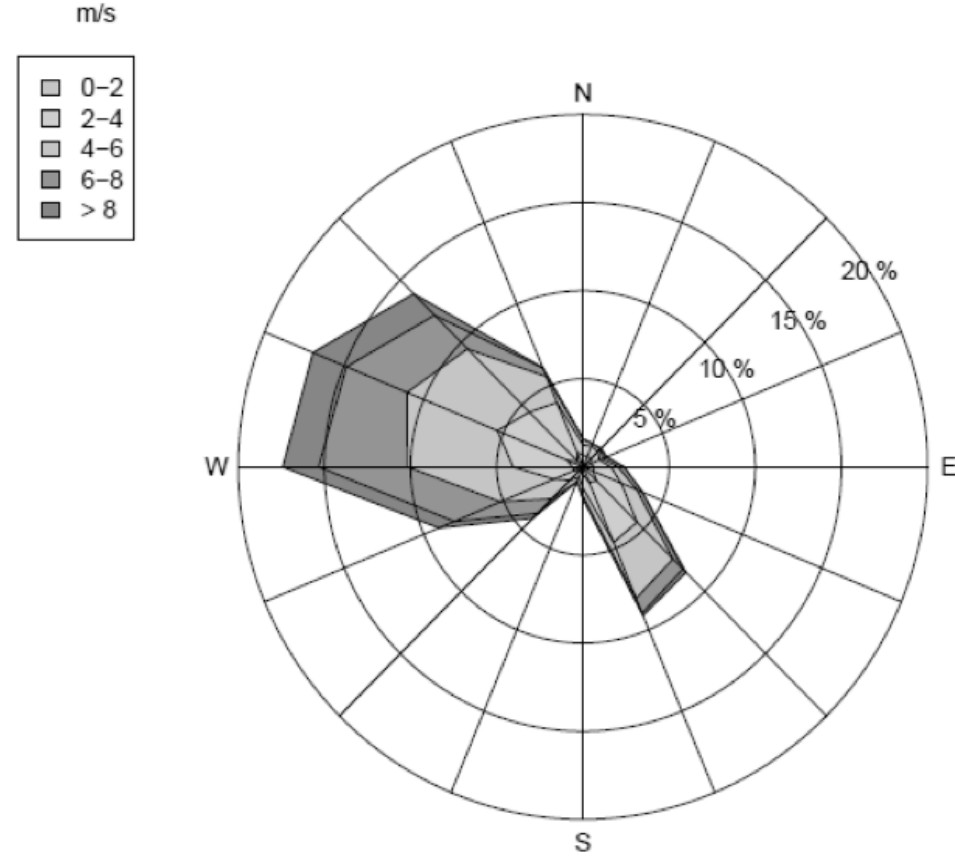

13    Figure 2. Distribution of the wind directions and wind speeds throughout the experimental

14    period (December 2012 – June 2013), as recorded by the meteorological station.





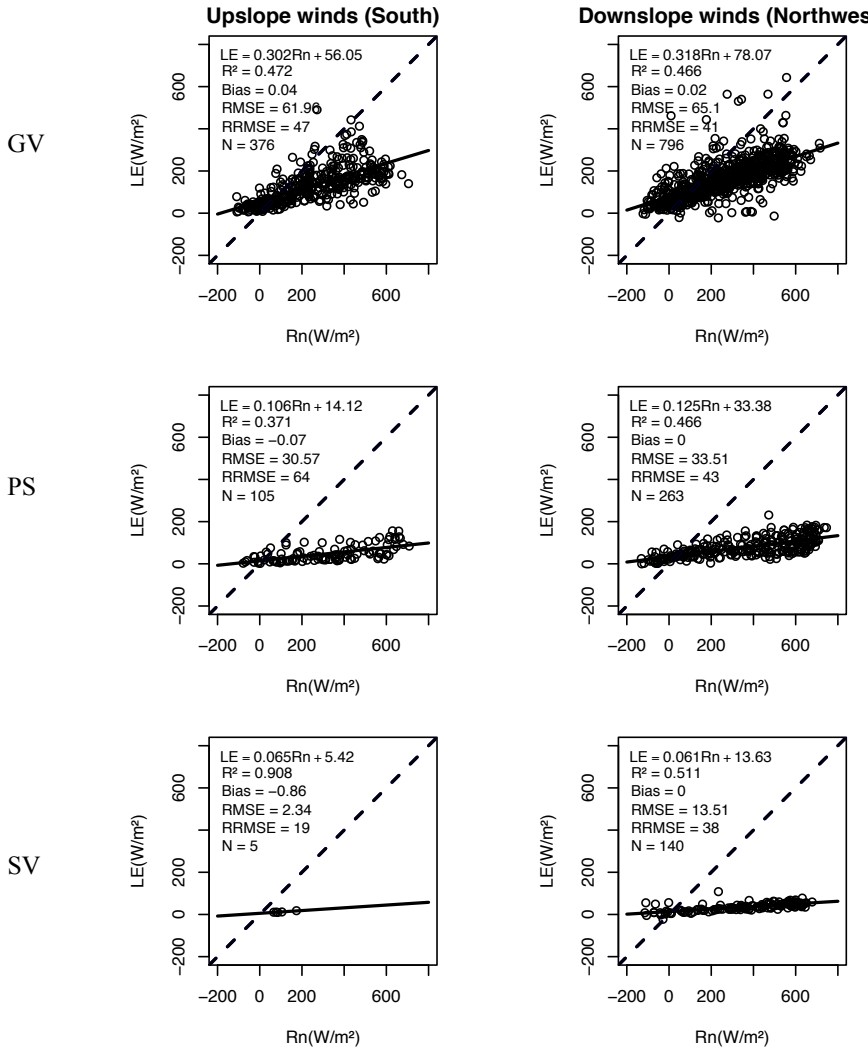

Figure 3. Calibration of the LE - Rn gap-filling method on field A. Columns 1 and 2 correspond to upslope and downslope winds, respectively. Lines 1, 2 and 3 correspond to the three periods (GV, PS, SV) that differed in vegetation phenology, soil water content and climatic conditions. The dashed line is the 1:1 line, and the continuous line is the regression line. $R^2$ is coefficient of determination. RMSE and RRMSE are absolute and relative root mean square errors, respectively. N is the number of flux data calculated over 30 min intervals.



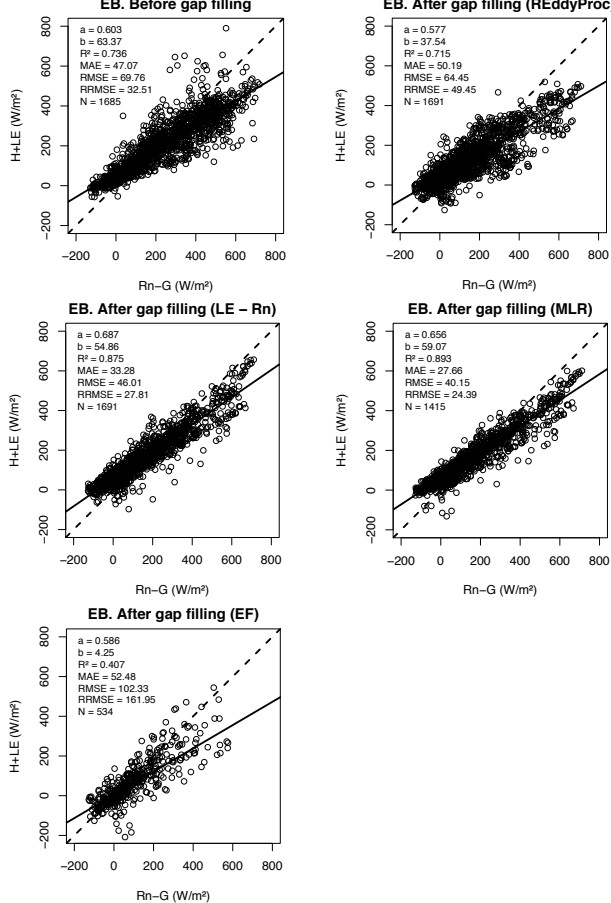

Figure 4. Energy balance closure (EB) for field A. Flux data are calculated over 30 minutes

intervals. Statistical indicators correspond to the comparison of convective energy (H + LE)

on y-axis against the available energy (Rn −G) on x-axis, before (top left subplot) and after

(other subplots) reconstruction of LE data by the four gap-filling methods. The dashed line is

the 1:1 line, and the continuous line is the regression line. Letters a and b are the slope and the

intercept of the linear regression, respectively. $R^2$ is coefficient of determination. MAE is the

mean absolute error. RMSE and RRMSE are absolute and relative root mean square errors,

respectively. N is the number of 30 min intervals data.



**List of Tables**
Table 1. Summary of relevant studies that deals with the performances of different gap filling methods for LE time series. Landform
classification includes flat / mountainous / hilly. Dataset splits are based on time window or on specific regimes. NR stands for "not reported".

| Study reference | Datasets or site | • Locations / landform / vegetation<br>• Boundary layer conditions / wind regimes<br>• Dataset split | Gap filling methods | Key results |
|---|---|---|---|---|
| Falge et al. (2001b) | EUROFLUX and AmeriFlux | • 18 sites / mountainous / four vegetation groups (conifers, deciduous forests, crops, grassland)<br>• NR / NR<br>• Time window (15 days) | Mean Diurnal Variation (MDV)<br>Look-Up Tables (LUT) | Good gap-filling performances. The two methods performed similarly. MDV estimates slight overestimated by LUT ones. |
| Cleverly et al. (2002) | Sevilleta and Bosque del Apache NWR | • 2 sites / NR / woody species<br>• Extremely stable / NR<br>• Time window (daily basis) | LE = a Rn + b<br>b significantly different from 0 | NR |
| Hui et al. (2004) | AmeriFlux | • 3 sites / hilly topography / forest (deciduous, coniferous, subalpine)<br>• NR / NR<br>• No split (1-year dataset) | Imputation methods | Good gap-filling performances. The methods performed similarly, multiple imputation method is easily portable in the context of worldwide networks. |
| Alavi et al. (2006) | NR | • 1 site / flat / winter wheat<br>• NR / NR<br>• Time windows according to the used method (1 year / 4-15 days / ±10 days / ±10 days) | Kalman filter<br>Multiple imputation (MI)<br>Mean Diurnal Variation (MDV)<br>Multiple regressions | Good gap-filling performances of Kalman filtering approach with smaller errors than the other methods. |
| Roupsard et al. (2006) | NR | • 1 site / flat / coconut plantation<br>• NR / NR<br>• Time window (1 month) | LE = a Rn + b<br>MDV to gap filling H | NR |
| Beringer et al. (2007) | Fluxnet | • 1 site / flat / woodland and open forest savanna<br>• NR / NR<br>• NR | Feed-forward back propagation (BPN) artificial neural network (ANN) (Papale et Valentini, 2003) | NR |

(continued on next page)



Table 1 (continued).

| Study reference | Datasets or site | • Locations / landform / vegetation<br>• Boundary layer conditions / wind regimes<br>• Dataset split | Gap filling methods | Key results |
|---|---|---|---|---|
| Zitouna-Chebbi (2009) | ORE OMERE | • 3 fields / hilly / winter cereals<br>• Neutrality or low instability / externally driven<br>• Based on upslope / downslope airflows (1-year dataset) | LE = a Rn + b | Good gap-filling performances when discriminating between the two prevailing wind directions. |
| Abudu et al. (2010) | Bosque del Apache NWR | • 1 site / NR / Salt cedar trees<br>• NR / NR<br>• Random split for calibration / testing over X% of existing data with X = [5, 10, 20, 30, 40]. | Feed-forward (FF) artificial neural networks (ANN) with different inputs | Best performance with the following inputs: daily maximum and minimum temperature, daily solar radiation, day of the year. |
| Chen et al. (2012) | NR | • 1 site / mountainous / forest (evergreens and hardwoods)<br>• NR / NR<br>• Based on nighttime and daytime (2-year dataset) | Two multivariate methods:<br>  • Multiple regressions (MRS)<br>  • K-nearest neighbors (KNNs) | KNN performed better than MRS |
| Eamus et al. (2013) | NR | • 1 site / flat plain / Savanna woodland<br>• Nocturnal stability / NR<br>• 10-day windows | Self-organizing linear output (SOLO) artificial neural network (ANN) | NR |
| This study | ORE OMERE | • 3 fields / hilly / winter cereals<br>• Neutrality or low instability / externally driven<br>• Based on vegetation phenology, and upslope / downslope airflows (1 crop growth cycle) | REddyProc (MDV / LUT based)<br>Linear regression method (LE - Rn)<br>Multiple linear regression (MLR)<br>Evaporative fraction (EF) | See results and discussion in the current paper |



Table 2. Details about experimental setup for each of the three flux stations: type of sensors used, acquisition and storage frequencies, and accuracies as given by manufacturers. HR and T stand for air relative humidity and temperature. Variables ux, uy and uz stand for 3D components of wind speed.

| Instrument type | Field A | Field B | Field C | Acquisition frequency | Storage frequency | Accuracy |
|---|---|---|---|---|---|---|
| Data logger | CR 3000 (Campbell Scientific Inc., USA) | | | | | |
| Sonic anemometer | CSAT3 (Campbell Scientific Inc., USA) | | Young-81000V (R.M Young, USA) | 20 Hz | 20 Hz | CSAT3: 0.001 m s$^{-1}$ Young: ±0.05 m s$^{-1}$ |
| Krypton hygrometer | KH20 (Campbell Scientific Inc., USA) | | | 20 Hz | 20 Hz | Unavailable |
| Thermo-hygrometer probe | HMP45C (Vaisala, Finland) | | | 1s | 15 mn | HR: ±1% T: ±0.2 °C |
| Net radiometer | NR01 (Hukseflux, Netherlands) | | | 1s | 15 mn | ±10% |
| Soil heat flux sensors | HFP (Hukseflux, Netherlands) (three per field) | | | 1s | 15 mn | -15% to +5% |

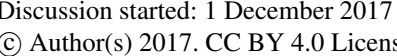


Table 3. Summary of the available latent heat flux (LE) data derived from the eddy
covariance measurements conducted on each of the three fields: dates of the beginning and
ending of measurement periods, number of total daytime data over 30 minutes intervals for
calculating the fluxes, numbers and percentages of data belonging to quality control classes
(I-IV: good quality data, V: rejected data), numbers and percentages of the missing data due
to dysfunctions of the KH20 sensors, and numbers and percentages of missing flux data due
to total shutdowns of the flux stations.

| Field | Beginning date | Ending date | Number of daytime 30 min intervals data | I-IV (%) | V (%) | KH20 dysfunctions (%) | System failure (%) |
|---|---|---|---|---|---|---|---|
| A | 06/ Dec /2012 | 11/ Jun /2013 | 4108 | 1685 (41) | 162 (4) | 1529 (37) | 732 (18) |
| B | 11/ Dec /2012 | 11/ Jun /2013 | 4007 | 820 (21) | 95 (2) | 2965 (74) | 127 (3) |
| C | 03/Jan/2013 | 11/ Jun /2013 | 3603 | 2198 (61) | 86 (2) | 616 (17) | 703 (20) |







Table 4. Splitting of the dataset into three periods when implementing the LE - Rn and MLR
gap filling methods. The three periods are labelled green vegetation (GV), pre-senescence
(PS) and senescent vegetation (SV). They are indicated along with the vegetation and climatic
conditions. LAI stands for green leaf area index, $ET_0$ stands for the reference
evapotranspiration. Minimum and maximum LAI values are averaged values over the three
fields A, B and C. Cumulative precipitation, mean $ET_0$ and mean air temperature are derived
from measurements at the meteorological station.

| Period | Dates | Main phenological stage | LAI min (m² / m²) | LAI max (m² / m²) | Cumulative precipitation (mm) | Mean $ET_0$ (mm / day) | Mean air temperature (°C) |
|---|---|---|---|---|---|---|---|
| GV | 06/Dec/2012 to 06/May/2013 | Seeding to beginning of dough stage | 0.07 | 2.37 | 357.5 | 2.6 | 11.4 |
| PS | 06/May/2013 to 28/May/2013 | Beginning of dough stage to fully ripened grain | 0.07 | 0.14 | 5 | 5.0 | 15.7 |
| SV | 28/May/2013 to 11/Jun/2013 | Fully ripened grain to senescence | - | - | 1 | 5.6 | 18.2 |





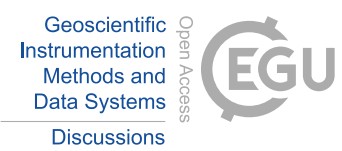

Table 5. Filling rate performance for each of the four gap-filling methods (REddyProc, LE - Rn, MLR, EF), expressed as the number and
percentage of reconstructible LE data that were actually reconstructed. The filling rates are given for each field (A, B, C) and each wind
direction. S and NW stand for south and northwest winds, respectively. Up and Down stand for upslope and downslope winds, respectively, for
the sloping fields A and B (field C was flat).

| | Field A (All data) | Field B (All data) | Field C (All data) | Field A S (Up) | Field A NW (Down) | Field A Total | Field B S (Down) | Field B NW (Up) | Field B Total | Field C S | Field C NW | Field C Total |
|---|---|---|---|---|---|---|---|---|---|---|---|---|
| Number of reconstructible data | 1691 | 2083 | 702 | 585 | 1106 | 1691 | 716 | 1367 | 2083 | 230 | 472 | 702 |
| **Number of reconstructed data (%)** | | | | | | | | | | | | |
| REddyProc | 1691 (100) | 2083 (100) | 702 (100) | 585 (100) | 1106 (100) | 1691 (100) | 716 (100) | 1367 (100) | 2083 (100) | 230 (100) | 472 (100) | 702 (100) |
| LE-Rn | 1691 (100) | 2083 (100) | 702 (100) | 585 (100) | 1106 (100) | 1691 (100) | 716 (100) | 1367 (100) | 2083 (100) | 230 (100) | 472 (100) | 702 (100) |
| MLR | 1415 (84) | 1789 (86) | 631 (90) | 489 (84) | 926 (84) | 1415 (84) | 626 (87) | 1163 (85) | 1789 (86) | 199 (87) | 432 (92) | 631 (90) |
| EF | 534 (32) | 398 (19) | 494 (70) | 226 (39) | 308 (28) | 534 (32) | 123 (17) | 275 (20) | 398 (19) | 156 (68) | 338 (72) | 494 (70) |




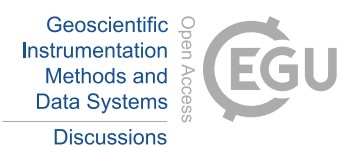

Table 6. Accuracy of LE retrievals for the four gap-filling methods (REddyProc, LE - Rn, MLR, EF). Fluxes were calculated over 30-min interval. Retrieval accuracy is given for each field (A, B, C) and each wind direction (NW and S stands for northwest and south winds, respectively) along with the corresponding airflow inclination when applicable (Up and Down stands for upslope and downslope winds, respectively). Accuracy is quantified using statistical indicators (absolute and relative RMSE, Bias, coefficient of determination R$^2$).

| | | Field A | Field B | Field C | Field A | | Field B | | Field C | |
|---|---|---|---|---|---|---|---|---|---|---|
| | | All data | All data | All data | S (Up) | NW (Down) | S (Down) | NW (Up) | S | NW |
| RMSE (W/m²) | REddyProc | 44.8 | 70.5 | 51.9 | 42.3 | 41.4 | 23.3 | 77.2 | 51.1 | 49.1 |
| | LE-Rn | 56.8 | 80.2 | 61.0 | 56.3 | 55.5 | 38.6 | 86.7 | 66.2 | 57.5 |
| | MLR | 58.3 | 61.7 | 59.7 | 55.1 | 55.8 | 37.3 | 61.9 | 61.9 | 57.0 |
| | EF | 57.5 | 87.3 | 62.8 | 48.1 | 56.8 | 42.9 | 98.2 | 63.8 | 57.8 |
| RRMSE (%) | REddyProc | 36 | 57 | 34 | 37 | 32 | 28 | 56 | 42 | 30 |
| | LE-Rn | 46 | 65 | 40 | 50 | 44 | 47 | 63 | 50 | 35 |
| | MLR | 45 | 48 | 37 | 47 | 41 | 45 | 43 | 45 | 34 |
| | EF | 47 | 70 | 41 | 43 | 44 | 52 | 70 | 48 | 36 |
| Bias (W/m²) | REddyProc | -1.34 | -1.13 | -0.65 | -2.14 | -0.90 | -0.96 | -1.58 | 2.20 | -0.80 |
| | LE-Rn | 0.01 | 0.00 | 0.00 | 0.00 | 0.01 | -0.03 | 0.00 | 0.01 | 0.00 |
| | MLR | 0.04 | -0.02 | 0.01 | -0.09 | 0.08 | -0.15 | 0.00 | 0.00 | 0.03 |
| | EF | -16.15 | -6.48 | -15.79 | -10.54 | -19.04 | -0.93 | -8.43 | -12.84 | -17.73 |
| R² | REddyProc | 0.74 | 0.42 | 0.78 | 0.75 | 0.78 | 0.75 | 0.40 | 0.83 | 0.81 |
| | LE-Rn | 0.58 | 0.25 | 0.69 | 0.56 | 0.61 | 0.32 | 0.25 | 0.59 | 0.73 |
| | MLR | 0.58 | 0.35 | 0.72 | 0.59 | 0.62 | 0.36 | 0.38 | 0.65 | 0.75 |
| | EF | 0.69 | 0.29 | 0.74 | 0.75 | 0.71 | 0.52 | 0.24 | 0.83 | 0.80 |