# Peer review of "Evaluating four gap-filling methods for eddy covariance measurements of evapotranspiration over hilly crop fields"

_Geoscientific Instrumentation, Methods and Data Systems, 2017_

## Referee Comment (RC1) · Anonymous Referee #1 · 20 Feb 2018

-General I would like to thank the authors for such an interesting article. Gap filling is very complex issue, and in this paper the authors show a complete and comprehensive evaluation of different gap-filling methods for Eddy covariance measurements over hilly crop fields.

-Specific The ratio of missing data reported in this work are much larger than those reported in former studies, why?, what are the reasons?

-Technical corrections Line 22: latent heat flux LEI, parenthesses? Line 32: net radiation Rn, parenthesses? Line 33: evaporative fraction EF, parenthesses? Line 50: latent heat flux LE, again and many times along the manuscript Line 333: Green LAI?,

is this correct?. LAI is leaf are index green or not green Line 378: same as 370

---

## Referee Comment (RC2) · Anonymous Referee #2 · 13 Mar 2018

Dear Nissaf Boudhina and co-authors, Dear Marina,

Please find below my review on the article "Evaluating four gap-filled methods for eddy covariance measurements of evaporation over hilly crop fields", submitted to Geoscientific Instrumentation Methods and Data Systems.

The authors provide a comparison and evaluation of four commonly used gap-filling methods for latent heat flux in hilly terrain. The basis form three short-term (Dec. 2012 to June 2013) eddy covariance (EC) measurements experiment realized in northeastern Tunesia. Following the quality control, the gap-filled LE estimates from the REddyProc method, linear regression, multi-linear regression and evaporative fraction are compared with measured values from the EC stations. Based on the results, the authors conclude that the performances and accuracies of the methods are comparable to instrumental accuracies.

The article is well written and the measurements, processing steps and evaluation have been formulated clearly. The argumentation is scientifically solid, as far as I can tell. The article does not say something new and unexpected, but provides additional facts and increments previous studies, which can be relevant to other studies working with EC measurements. However, the article is very long and requires a strong stamina of the reader. Although the introduction is well written, the reader already needs some patient here. Scientists working in this field, and those you are probably addressing, should be familiar the basic concepts. Rather than quickly remind the readers of the variety of methods and the problem they already know, they have to be patient until the story narrows and gets to the part where the larger problem and knowledge gap, which you propose to answer, is specified. The article, and in particular the introduction, should be shortened in order to minimize the risk to loose the reader on the way. Perhaps, it might be a good idea to put some of the details in the supplement.

Here are some suggestions, where the article can be shortened and therefore becomes more accessible to the reader:

(i)     L68-88 discusses the basics of common gap-filling methods and provides a brief description of the method. This paragraph can be shortened without loss of the overall story, since the used methods are introduced in Section 3. The most important part of this paragraph is the last sentences, which is basically the motivation for this study: Different gap-filling methods have not been evaluated so far in hilly topography.

(ii)    L90-97 talks about hilly watersheds and the urgent need to understand the evaporation process. The study discusses the accuracy of gap-filling methods and does not address the understanding. Therefore, the paragraph is not relevant for this work and can be removed.

(iii)   L98-116: This paragraph can be shortened by briefly mentioning how terrain complexity and airflow characteristics impacts evapotranspiration, e.g. terrain complexity dictates the radiation fluxes, stability and flow dynamics and hence differs for hilly and alpine terrain.

(iv)    Section 2.5.2 discusses two commonly used coordinate rotation algorithms. Since the planar fit is not used at all in this study, the detail of this algorithm can be removed.

(v)     L470-479: These lines repeat the method constraints which were already discussed in the methods section.

(vi)    While the results section is normally written in past tense, I think that the facts in the discussion section should be written in present tense, e.g. (L622) REddyProc relies on existing … or (L651) This emphasizes the impact …

**Specific comments:**

L46-48: The opening offers no direction as to where the story is going. It geos over evapotranspiration, biomass production, photosynthesis, surface energy balance, water balance, Mediterranean climate, and managing agricultural activities. The first paragraph should set the direction of the paper. More precisely it should identify the problem that drives the research and target the audience. I suggest to rewrite the first paragraph to make the manuscript more interesting to the reader.

L50: Please introduce the abbreviation "… latent heat flux (LE)" and use the abbreviation throughout the text, e.g. L56, L72 etc.

L56: Why do environmental sciences require hourly evapotranspiration measurements?

L62: Isn't the expression "dysfunction" used for medical disorder?

L99: Better use "characteristics" instead of "specificities".

L102: Please write "… topographic characteristics and ABL conditions differ between hilly areas and mountainous terrain."

L104; L105; L108: Please avoid the repetition of "Regarding …".

L105/106: What do you want to say with "…, hilly areas rise over small fractions of the daytime ABL, and the overlying airflows are slightly influenced by stratification"?

L107: Change "instable" to "unstable".

L108: I think you refer as "externally wind" the "dynamically induced winds"?

L109: Dynamically induced winds can be very important for alpine terrain. Sometimes these winds superimpose thermal winds and the interaction of topography with the wind field results in lee rotors and flow splitting. So I don't agree that these winds are more frequent than in complex terrain.

L111-116: Please split the long sentence into several small sentences.

L133: Please give the altitudes of the experimental sites.

L143: "…, yearly precipitation sums …"

L144: Are the numbers correct? Is there more evapotranspiration than precipitation? Later in the text (L150) you mention that the agriculture is rainfed.

L194/195: Better write: "…, since the sensible and latent heat fluxes are insignificant (small) during night time."

L200: What means "normal to local topography"?

L204: Please introduce the abbreviation DEM.

L220: "H and LE fluxes were averaged over 30 minute intervals."

L261: I think the ST assesses the stationarity and not the homogeneity of turbulence.

L262: Does the ITC really test the spatial homogeneity or the isotropy of turbulence?

L272-279: What impact has the ridge on the measurements of site A and B? Do you observe eddies during strong large-scale winds?

L285: Remove "at daily timscale".

L287-288: the sentence "This value …" is neither relevant nor representative for this study and should be removed.

L289-290: Why is there no influence of the topography on the wind field?

L294/295: As far as I see, the secondary maximum of the wind is between 120° and 160° and not between 70° and 220°.

L301: Please give the units for all variables, e.g. z and D.

L304: Remove "notably".

L314: Please write "land-sea breeze".

L329: Please change "was" to "were".

L330: Please rewrite "… the experiment was typified …".

L335/336: The LAI is not necessary in this case and can be removed.

L350: Please specify the ratio, e.g. the ratio of the original to filtered time series …

L358/361: Remove the blank between the number and %.

L436: Remove 'Evaporative fraction' and 'latent heat flux'.

L450-456: This paragraph should be moved to the Results section.

L587: The expression 'The method performances could be either different or similar before and after splitting …' is trivial and can be removed.

L637: Better write: 'Overall, the four methods were able to fill all gaps in the time series, …'.

L669: Please write 'The EF method provided lower accuracies'.

L683: Change 'could be' to 'are'.

Figure 4: It would be interesting to provide the same figure (maybe in the supplements) for site B.

Table 6: I think the data is more accessible to the reader when presented with a box plot.

---

## Author Comment (AC2) · 10 Apr 2018

**R1C1. General I would like to thank the authors for such an interesting article. Gap filling is very complex issue, and in this paper the authors show a complete and comprehensive evaluation of different gap-filling methods for Eddy covariance measurements over hilly crop fields.**

Reply to R1C1. We highly appreciate your positive comment on our work.

No revision for R1C1.
* * *
**R1C2. Specific The ratio of missing data reported in this work are much larger than those reported in former studies, why?, what are the reasons?**

Reply to R1C2. This was explained in the submitted manuscript, third paragraph of Section 2.7: "The low ratio we obtained in the current study was ascribed to KH20 dysfunctions and total shutdown of flux stations. Furthermore, the KH20 sensor installed on field B was out of order from the end of March until the end of the experiment, because of severe instrumental dysfunctions."

No revision for R1C2.
* * *
**R1C3. Technical corrections Line 22: latent heat flux LEl, parentheses? Line 32: net radiation Rn, parentheses? Line 33: evaporative fraction EF, parentheses? Line 50: latent heat flux LE, again and many times along the manuscript Line 333: Green LAI?, is this correct?. LAI is leaf are index green or not green Line 378: same as 370**

Reply to R1C3. Several items to be considered, listed below.
- We checked and corrected across the whole manuscript for each mentioned acronym: EC, LE, Rn, G, H, EF, MLR, LUT, MDV, ABL, ASL, DEM, ST, ITC, LAI (now GAI), GV, PS, SV, LOOCV.
- To avoid any confusion, we defined leaf area index of green leaves as GAI instead of LAI.

Revisions for R1C3.

- Corrections on acronyms
  - Eddy covariance (EC) technique allows continuous measurements of latent heat flux (LE).
  - Indeed, evapotranspiration (or latent heat flux LE)…
  - Table 1 summarizes the few studies that addressed measurements of  LE…
  - In the context of obtaining continuous time series of evapotranspiration from EC measurements of  LE…
  - It extends from the Jebel Abderrahmane to the Korba Laguna, and it includes the Kamech watershed (outlet at 36°52'30"N, 10°52'30"E, 108 m above sea level - asl -) that has an area of 2.7 × 0.9 km² (Figure 1).

- On fields A, B and C, each flux station collected measurements of the land surface energy fluxes: net radiation (Rn), soil heat flux (G), sensible (H) and latent heat (LE) fluxes.
- For each flux station,  was estimated by averaging the measurements collected with the three soil heat flux sensors.
- H and LE were calculated from the 20 Hz data collected by the sonic anemometers and the krypton hygrometers…
- Table 3 displays the amounts of available data derived from EC measurements over the three fields, when considering LE.
- For  H, the percentages of data belonging…
- Evaporative fraction (EF) is defined as the ratio of  LE over available energy (Rn - G)…
- We quantified the retrieval accuracies of the four gap-filling methods by comparing reference data and gap-filling retrievals of  LE over…
- Overall, the RMSE values between reference data and gap-filling retrievals of  LE ranged between…
- …we selected the commonly used REddyProc method that relies on Look Up Tables (LUT) and Mean Diurnal Variation (MDV) to fill missing flux data…
- …both derived from a four-meter spatial resolution digital elevation model (DEM) obtained with…
- Leaf area index of green leaves (GAI) was measured using a planimeter.
-  GAI reached its maximum on April 11…

- Corrections on LAI
  - Leaf area index of green leaves (GAI) was measured using a planimeter. Every two weeks, all leaves were collected within three one-meter-long transects to derive a spatially averaged value.  GAI reached its maximum on April 11, and maximum values were 2.5 m²/m², 2.3 m²/m² and 2.3 m²/m² for fields A, B and C respectively.

o Table 4. Splitting of the dataset into three periods when implementing the LE - Rn and MLR gap filling methods. The three periods are labelled green vegetation (GV), pre-senescence (PS) and senescent vegetation (SV). They are indicated along with the vegetation and climatic conditions.  GAI stands for  leaf area index of green leaves, $ET_0$ stands for the reference evapotranspiration. Minimum and maximum  GAI values are averaged values over the three fields A, B and C. Cumulative precipitation, mean $ET_0$ and mean air temperature are derived from measurements at the meteorological station.

| Period | Dates | Main phenological stage |  GAI min (m² / m²) |  GAI max (m² / m²) | Cumulative precipitation (mm) | Mean $ET_0$ (mm / day) | Mean air temperature (°C) |
|---|---|---|---|---|---|---|---|
| GV | 06/Dec/2012 to 06/May/2013 | Seeding to beginning of dough stage | 0.07 | 2.37 | 357.5 | 2.6 | 11.4 |
| PS | 06/May/2013 to 28/May/2013 | Beginning of dough stage to fully ripened grain | 0.07 | 0.14 | 5 | 5.0 | 15.7 |
| SV | 28/May/2013 to 11/Jun/2013 | Fully ripened grain to senescence | - | - | 1 | 5.6 | 18.2 |

**Reply to Reviewer #2 comments**

**Dear Nissaf Boudhina and co-authors, Dear Marina,**
**Please find below my review on the article "Evaluating four gap-filled methods for eddy covariance measurements of evaporation over hilly crop fields", submitted to Geoscientific Instrumentation Methods and Data Systems.**
**The authors provide a comparison and evaluation of four commonly used gap-filling methods for latent heat flux in hilly terrain. The basis form three short-term (Dec. 2012 to June 2013) eddy covariance (EC) measurements experiment realized in northeastern Tunesia. Following the quality control, the gap-filled LE estimates from the REddyProc method, linear regression, multi-linear regression and evaporative fraction are compared with measured values from the EC stations. Based on the results, the authors conclude that the performances and accuracies of the methods are comparable to instrumental accuracies.**

**R2C1. The article is well written and the measurements, processing steps and evaluation have been formulated clearly. The argumentation is scientifically solid, as far as I can**

**tell. The article does not say something new and unexpected, but provides additional facts and increments previous studies, which can be relevant to other studies working with EC measurements. However, the article is very long and requires a strong stamina of the reader. Although the introduction is well written, the reader already needs some patient here. Scientists working in this field, and those you are probably addressing, should be familiar the basic concepts. Rather than quickly remind the readers of the variety of methods and the problem they already know, they have to be patient until the story narrows and gets to the part where the larger problem and knowledge gap, which you propose to answer, is specified. The article, and in particular the introduction, should be shortened in order to minimize the risk to loose the reader on the way. Perhaps, it might be a good idea to put some of the details in the supplement. Here are some suggestions, where the article can be shortened and therefore becomes more accessible to the reader:**

Reply to R2C1. We highly appreciate your positive comment on our work, and are grateful to you for the whole set of suggestions that helped us to greatly improve the manuscript. We account for about 95% of your comments, and we modified the manuscript accordingly, as detailed hereafter. When some comments were not taken into account, we explained why.

No revision for R2C1.
* * *
**R2C2. (i) L 68-88 discusses the basics of common gap-filling methods and provides a brief description of the method. This paragraph can be shortened without loss of the overall story, since the used methods are introduced in Section 3. The most important part of this paragraph is the last sentences, which is basically the motivation for this study: Different gap-filling methods have not been evaluated so far in hilly topography.**

Reply to R2C2. We removed the references that are mentioned in Table 1, and we removed details about gap-filling methods. We kept the two main conclusions: (1) time series are often split before applying gap-filling, and (2) different gap-filling methods have not been evaluated so far in hilly topography.

Revisions for R2C2.

Most existing gap-filling methods  are devoted to carbon dioxide ($CO_2$) measurements (Aubinet et al. 1999; Falge et al. 2001a; Goulden et al. 1996; Greco and Baldocchi 1996; Grünwald and Bernhofer 1999; Moffat et al. 2007; Reichstein et al. 2005; Ruppert et al. 2006). Table 1 summarizes the few studies that addressed measurements of  LE, along with underlying methodologies and resulting performances . ~~The most usual gap-filling methods are Look-Up Tables (LUT) based methods, Mean Diurnal Variation (MDV) method and multivariate approaches. LUT based methods consist in filling gaps with data collected under similar meteorological conditions. MDV based methods consist in replacing missing values by the mean obtained on adjacent days. Multivariate approaches (i.e., artificial neural networks, principle component analysis, interpolations and regressions) consist in filling gaps using linear or non-linear relationships that involve drivers of evapotranspiration such as meteorological variables, soil water content or net radiation.~~ Prior to gap filling, time series are often split in different ways

according to the experimental conditions (e.g., nighttime / daytime, wind directions, vegetation phenology, weekly or monthly time windows), so that missing data are filled with observations collected in similar conditions for micrometeorology, vegetation phenology and water status. Overall, gap-filling methods for LE time series have been evaluated over flat, hilly and mountainous areas. However, the existing studies for hilly areas did not address their specific conditions (Hui et al. 2004), or  were restricted  to one gap-filling method  (Zitouna-Chebbi 2009).
* * *
**R2C3. (ii) L 90-97 talks about hilly watersheds and the urgent need to understand the evaporation process. The study discusses the accuracy of gap-filling methods and does not address the understanding. Therefore, the paragraph is not relevant for this work and can be removed.**

Reply to R2C3. We did not remove this paragraph that motivates our study. However, we removed the confusing sentence pointed out by the comment. Also, we moved this paragraph at the beginning of Introduction, to focus on key issues on which relies the current study, in accordance with R2C8.

Revisions for R2C3.

——————Hilly watersheds are widespread within coastal areas around the Mediterranean basin, as well as in Eastern Africa, India and China. They experience agricultural intensification since hilly topographies allow water-harvesting techniques that compensate for precipitation shortage (Mekki et al. 2006). Their fragility is likely to increase with climate change and human pressure, especially as water scarcity already limits crop production. In this context, understanding evapotranspiration processes within hilly watersheds is paramount for the design of decision support tools devoted to water resource management (McVicar et al. 2007). Indeed, evapotranspiration (or latent heat flux LE)  directly drives biomass production through intertwining with  photosynthesis  (Olioso et al. 2005) and  it is a major term of water balance,  up to 2/3 of the annual water balance for semi-arid  / subhumid Mediterranean climates (Montes et al. 2014; Moussa et al. 2007; Yang et al. 2014).
* * *
**R2C4. (iii) L 98-116: This paragraph can be shortened by briefly mentioning how terrain complexity and airflow characteristics impacts evapotranspiration, e.g. terrain complexity dictates the radiation fluxes, stability and flow dynamics and hence differs for hilly and alpine terrain.**

Reply to R2C4. We reduced in accordance to reviewer comments. Finally, we kept the last sentence that motivates the need for tailoring gap-filling methods when addressing hilly areas.

Revisions for R2C4.

Gap-filling methods for LE have to be designed in accordance with the terrain  characteristics that impact evapotranspiration. Conversely to flat terrains that correspond to slope lower than 2% (Appels et al. 2016), solar and net radiations within sloping terrains change depending on slope orientation, with larger values for ecliptic-facing slopes (Holst et al. 2005). Over sloping terrains, the conditions of topography, atmospheric stability and airflow within the atmospheric boundary layer (ABL)  differ between hilly areas  and mountainous areas (Dupont et al. 2008; Hammerle et al. 2007; Hiller et al. 2008; Prima et al. 2006; Raupach and Finnigan 1997). ~~Regarding topography, hilly areas depict lower slopes on average, and Prima et al. (2006) proposed a threshold value of 22%. Regarding atmospheric stability, hilly areas rise over small fractions of the daytime ABL, and the overlying airflows are slightly influenced by stratification, which corresponds to neutral or instable conditions (Raupach and Finnigan 1997). Regarding wind regimes, externally driven winds are more frequent within hilly areas, as compared to mountainous areas with anabatic and katabatic flows (Hammerle et al. 2007; Hiller et al. 2008), and wind regimes differ much between the upwind and lee sides of hills (Dupont et al. 2008; Raupach and Finnigan 1997). Therefore, theon which rely the, mostly,within hilly areas,~~ because of changes in airflow inclination (Zitouna-Chebbi et al. 2012; 2015), and therefore changes in aerodynamic properties (Blyth 1999; Rana et al. 2007).
* * *
**R2C5. (iv) Section 2.5.2 discusses two commonly used coordinate rotation algorithms. Since the planar fit is not used at all in this study, the detail of this algorithm can be removed.**

Reply to R2C5. We corrected in accordance to reviewer comments.

**2.5.2. Coordinate rotations**
When calculating energy fluxes with the EC method, it is conventional to rotate the coordinate system of the sonic anemometer (Kaimal and Finnigan 1994). Coordinate rotations were originally designed to correct the vertical alignment of the sonic anemometer over flat terrains, and they are commonly used over non-flat terrains to virtually align the sonic anemometer perpendicularly to the mean airflow, in an idealized homogeneous flow.

The main potential drawback of the double rotation method is that a significant variability in rotation angles can be observed at low wind speeds (Turnipseed et al. 2003). Since  our study area was typified by large wind speeds (Zitouna-Chebbi et al. 2012; 2015), we selected the double rotation method  that is applied to each time interval over which the convective fluxes are calculated (30 minutes in our case). After  a first rotation (yaw angle) that cancels the lateral component of the horizontal wind speed , a second

rotation (pitch angle) is applied around a horizontal axis perpendicular to the main wind direction, to cancel the mean vertical wind speed. Thus, it implicitly accounts for changes in wind direction and vegetation height that are likely to be constant over 30-minute intervals.

~~Both double rotation and planar fit methods have advantages and drawbacks. On the one hand, a significant variability in rotation angles can be observed at low wind speeds with the double rotation method (Turnipseed et al. 2003). On the other hand, the planar fit method must be applied for different sectors of wind direction and for different intervals of vegetation height in case of sloping terrains and changes in vegetation height (Zitouna-Chebbi et al. 2012; 2015). Since our study area was typified by large wind speeds (Zitouna-Chebbi et al. 2012; 2015), we selected the double rotation method.~~

**R2C6. (v) L 470-479: These lines repeat the method constraints which were already discussed in the methods section.**

Reply to R2C6. We rewrote the two last paragraphs of Section 3.2 to remove redundancies.

Revisions for R2C6.

Most existing gap filling methods for LE measurements include a prior splitting of the time series to be filled (Table 1), so that missing data are filled with existing observations collected under similar conditions . REddyProc relies on time windows up to  280 days, and  the EF method relies on a daily time window .  For both the LE - Rn  and  MLR method that assumes  stable regression between energy fluxes,

 we split the dataset into three periods that differed in vegetation phenology. This led to  account for changes in soil water content and vegetation height at monthly to seasonal timescales.

**R2C7. (vi) While the results section is normally written in past tense, I think that the facts in the discussion section should be written in present tense, e.g. (L 622) REddyProc relies on existing ... or (L 651) This emphasizes the impact ...**

Reply to R2C7. We changed in accordance to reviewer comments.

Revisions for R2C7.

- Indeed, REddyProc  relies on existing LE values within a given time window, either corresponding to similar meteorological variables or derived from averaged diurnal courses.

- This  emphasizes the impact of changes in soil water content and vegetation canopy at monthly to seasonal timescales.
- The slightly better accuracies obtained with REddyProc  indicates that this method was able to find appropriate LE values under similar meteorological conditions or within a given time window, in spite of possible changes in soil water content. LE - Rn and MLR  provided very similar accuracies.
* * *
Specific comments:
* * *
**R2C8. L 46-48: The opening offers no direction as to where the story is going. It geos over evapotranspiration, biomass production, photosynthesis, surface energy balance, water balance, Mediterranean climate, and managing agricultural activities. The first paragraph should set the direction of the paper. More precisely it should identify the problem that drives the research and target the audience. I suggest to rewrite the first paragraph to make the manuscript more interesting to the reader.**

Reply to R2C8. Following reviewer comment, we removed this paragraph, and we replaced it by the fourth paragraph that deals with societal challenges for hilly watersheds. Please see our reply to R2C3.

Revisions for R2C8. Please see our revisions for R2C3.
* * *
**R2C9. L 50: Please introduce the abbreviation "... latent heat flux (LE)" and use the abbreviation throughout the text, e.g. L 56, L 72 etc.**

Reply to R2C9. Please see our reply to R1C3 about acronyms.

Revisions for R2C9. Please see our revisions for R1C3 about acronyms.
* * *
**R2C10. L 56: Why do environmental sciences require hourly evapotranspiration measurements?**

Reply to R2C10. Hourly evapotranspiration measurements are necessary in semi-arid because of water stress that induces stomatal closure in afternoon and therefore asymmetry between morning and noon transpiration. This is a key issue that drives the feature of the NASA ECOSTRESS mission. (http://adsabs.harvard.edu/abs/2015AGUFM.H31J..07F).

No revision for R2C10.
* * *
**R2C11. L 62: Isn't the expression "dysfunction" used for medical disorder?**

Reply to R2C11. Thanks for reporting this error. We replaced dysfunctions with malfunctions across the whole manuscript.

Revisions for R2C11.

- However, time series of EC measurements often experience large portions of missing data, because of instrumental  malfunctions or quality filtering.
- However, common time series of eddy covariance (EC) measurements, which are nowadays considered as the reference method, include missing data because of experimental troubles such as power failures or instrumental malfunctions.
- Missing LE data stemmed from (1) total shutdowns of flux stations, following battery discharges or vandalism acts; (2) malfunctions of KH20 krypton hygrometers after precipitation events when air humidity permeated the sensor because of seal degradation; and (3) rejection of LE data identified as class V data by ST and ITC tests (Section 2.5.3).
- It gives the beginning and ending dates of the EC measurements, the number of daytime data over 30 minutes intervals, the numbers and proportions of data with good (classes I to IV) and bad quality (class V) according to ST and ITC tests, the number of missing data due to malfunctions  of the Krypton hygrometer (KH20), and the number of missing data because of total shutdown of flux stations.
- The low ratio we obtained in the current study was ascribed to KH20 malfunctions  and total shutdown of flux stations.
- Furthermore, the KH20 sensor installed on field B was out of order from the end of March until the end of the experiment, because of severe instrumental malfunctions.
- Therefore, we addressed the filling of the gaps that resulted from malfunctions  of the KH20 sensors and quality filtering only.
- The filling performances were given in accordance with the number of reconstructible data (LE missing data because of both KH20 malfunctions  and quality filtering). They were expressed as the ratio of reconstructed to reconstructible data.
- For the three fields (A, B, C) and the two wind directions (northwest, south), Table 5 displays the number of reconstructible data (LE missing data because of KH20 malfunctions  or LE data belonging to quality class V), as well as the number and percentage of reconstructed data by the four methods (REddyProc, LE - Rn, MLR and EF).
- The total number of reconstructible data in Table 5 corresponds to that given in Table 3 (i.e. sum of LE missing data because of KH20 malfunctions  and of LE data belonging to quality class V), apart from field B (2083 versus 3060) for which we restricted the time period to the GV period, since no LE data were available on periods PS and SV because of the KH20 shutdown (second item in Section 3.3).
* * *
**R2C12. L 99: Better use "characteristics" instead of "specificities".**

Reply to R2C12. Corrected accordingly.

Revisions for R2C12.

Gap-filling methods for LE have to be designed in accordance with the terrain  characteristics that impact evapotranspiration.
* * *
**R2C13. L 102: Please write "... topographic characteristics and ABL conditions differ between hilly areas and mountainous terrain."**

Reply to R2C13. Corrected accordingly.

Revisions for R2C13.

Over sloping terrains, the conditions of topography, atmospheric stability and airflow within the atmospheric boundary layer (ABL) are very different for between hilly areas as compared to and mountainous areas (Dupont et al. 2008; Hammerle et al. 2007; Hiller et al. 2008; Prima et al. 2006; Raupach and Finnigan 1997).
* * *
**R2C14. L 104; L 105; L 108: Please avoid the repetition of "Regarding ...".**

Reply to R2C14. This part was removed in accordance to R2C4.

No revision for R2C14.
* * *
**R2C15. L 105/106: What do you want to say with "..., hilly areas rise over small fractions of the daytime ABL, and the overlying airflows are slightly influenced by stratification"?**

Reply to R2C15. This part was removed in accordance to R2C4.

No revision for R2C15.
* * *
**R2C16. L 107: Change "instable" to "unstable".**

Reply to R2C16. This part was removed in accordance to R2C4.

No revision for R2C16.
* * *
**R2C17. L 108: I think you refer as "externally wind" the "dynamically induced winds"?**

Reply to R2C17. This part was removed in accordance to R2C4.

No revision for R2C17.
* * *
**R2C18. L 109: Dynamically induced winds can be very important for alpine terrain. Sometimes these winds superimpose thermal winds and the interaction of topography with the wind field results in lee rotors and flow splitting. So I don't agree that these winds are more frequent than in complex terrain.**

Reply to R2C18. This part was removed in accordance to R2C4.

No revision for R2C18.
* * *
**R2C19. L 111-116: Please split the long sentence into several small sentences.**

Reply to R2C19. Corrected accordingly.

Revisions for R2C19.

Most  existing gap-filling methods rely on  co-variation of convective fluxes with meteorological variables, or temporal auto-correlation of the convective fluxes. Within hilly areas, these relationships  are likely to change with wind direction and vegetation development,  because of changes in airflow inclination (Zitouna-Chebbi et al. 2012; 2015), and therefore changes in aerodynamic properties (Blyth 1999; Rana et al. 2007).
* * *
**R2C20. L 133: Please give the altitudes of the experimental sites.**

Reply to R2C20. Mentioned in the submitted version (and still mentioned in the revised version). Second paragraph of Section 2.1: "Terrain elevation ranges from 94 m asl to 194 m asl, and terrain slopes range between 0% and 30%, the quartiles being 6%, 11% and 18% (Zitouna-Chebbi et al. 2012)."

No revision for R2C20.
* * *
**R2C21. L 143: "..., yearly precipitation sums ..."**

Reply to R2C21. Corrected accordingly.

Revisions for R2C21.

The climate of the Kamech watershed is sub-humid Mediterranean. Over the [1995-2014] period, cumulated values at the yearly timescale for precipitation and Penman-Monteith reference crop evapotranspiration (Allen et al. 1998) are 624 mm and 1526 mm, respectively.
* * *
**R2C22.** L 144: Are the numbers correct? Is there more evapotranspiration than precipitation? Later in the text (L 150) you mention that the agriculture is rainfed.

Reply to R2C22. Yes the number are correct. Indeed, cumulated value at the yearly timescale for reference evapotranspiration is larger than that for precipitation, even for rainfed agriculture. First, we deal with Penman Monteith (PM) reference crop evapotranspiration, which is representative of atmospheric demand and not of actual evapotranspiration. Thus, yearly PM reference evapotranspiration is larger than yearly precipitation, which is a characteristic of subhumid / semiarid climates. Second, rainfed agriculture is possible in such conditions since precipitation is concentrated during autumn / spring and crop growth cycles spread over the [October - May] period.

No revision for R2C22.
* * *
**R2C23. L 194/195: Better write: "..., since the sensible and latent heat fluxes are insignificant (small) during night time."**

Reply to R2C23. Corrected accordingly.

Revisions for R2C23.

All instruments were manufacturer-calibrated. Hereafter in the paper, we focused on daytime measurements, since latent heat flux is insignificant (small) during night time.
* * *
**R2C24. L 200: What means "normal to local topography"?**

Reply to R2C24. It means normal to topography[1], where slope and aspect are calculated from a 4 m-resolution DEM in the vicinity of Rn sensors.

No revision for R2C24.
* * *
**R2C25. L 204: Please introduce the abbreviation DEM.**

Reply to R2C25. Corrected accordingly.

Revisions for R2C25.

We characterized local topography with slope (topographical zenith with nadir as origin) and aspect (topographical azimuth with north as origin), both derived from a four-meter spatial resolution digital elevation model (DEM) obtained with a stereo pair of Ikonos images (Raclot and Albergel 2006).
* * *
**R2C26. L 220: "H and LE fluxes were averaged over 30 minute intervals."**

Reply to R2C26. We disagree. It is a covariance calculation, along with data processing for EC measurements, as detailed in Section 2.5.1 & 2.5.2.

No revision for R2C26.
* * *
**R2C27. L 261: I think the ST assesses the stationarity and not the homogeneity of turbulence.**

Reply to R2C27. Corrected accordingly
* * *
[1] https://www.sciencedirect.com/science/article/pii/S1463500317301117

Revisions for R2C27.

The ST test assesses the stationarity homogeneity of the turbulence over time, while the ITC test assesses the good development spatial homogeneity of the turbulence, both tests being performed over each 30-minute interval.
* * *
**R2C28. L 262: Does the ITC really test the spatial homogeneity or the isotropy of turbulence?**

Reply to R2C28. Corrected accordingly

Revisions for R2C28.

These tests verify that the theoretical requirements for the EC measurements are fulfilled. The ST test assesses the stationarity homogeneity of the turbulence over time, while the ITC test assesses the good development spatial homogeneity of the turbulence, both tests being performed over each 30-minute averaging interval.
* * *
**R2C29. L 272-279: What impact has the ridge on the measurements of site A and B? Do you observe eddies during strong large-scale winds?**

Reply to R2C29. It is first to be mentioned that the topography around sites A and B is rather gentle. We never observed eddies, even during strong large-scale winds that are common on our study site. Concerning site A, the impact of the topography on flux measurements has been described in Zitouna et al. (2012, 2015) and, since the ridge topography is quite symmetrical, we assume that ridge effect on site B is similar.

No Revisions for R2C29.
* * *
**R2C30. L 285: Remove "at daily timescale".**

Reply to R2C30. Corrected accordingly.

Revisions for R2C30.

Over the same period, the reference evapotranspiration $ET_0$ recorded by the meteorological station ranged between 1.1 and 5.8 mm day$^{-1}$ at the daily timescale, with a cumulated value of 510 mm.
* * *
**R2C31. L 287-288: the sentence "This value ..." is neither relevant nor representative for this study and should be removed.**

Reply to R2C31. Corrected accordingly.

Revisions for R2C31.

The wind speed value recorded during the experimental period by the meteorological station was 4 m s$^{-1}$ on average.  The averaged wind speed value recorded by the meteorological station was very close to those recorded by the sonic anemometers installed on the flux stations within field A, B and C, with differences lower than 0.4 m s$^{-1}$.
* * *
**R2C32. L 289-290: Why is there no influence of the topography on the wind field?**

Reply to R2C32. As mentioned in section 2.6.1, 2$^{nd}$ paragraph, a high spatial homogeneity of the horizontal wind speed was observed between the three measurements sites (A, B, C) and with the meteorological station, during this experiment. This homogeneity was also observed during several previous studies conducted on different locations on the same site (Zitouna-Chebbi, 2009). This might be related to the fact that we never conducted measurements on the steepest parts of the catchment, where some influence of the topography on the wind field would be likely to occur.

No Revision for R2C32.
* * *
**R2C33. L 294/295: As far as I see, the secondary maximum of the wind is between 120° and 160° and not between 70° and 220°.**

Reply to R2C33. Yes, but we need to split in two sectors so that (1) all data are included, (2) we have two sectors corresponding up / down winds. Additionally, increasing the sector number would lead to sectors with very few data. Also, we follow wind sector designed in Zitouna et al (2012, 2015) on the same study site.

No revision for R2C33.
* * *
**R2C34. L 301: Please give the units for all variables, e.g. z and D.**

Reply to R2C34. Corrected accordingly.

Revisions for R2C34.

Micrometeorological conditions were analyzed using the atmospheric stability parameter $\xi = (z\text{-}D) / L_{MO}$, where z is measurement height (in meters), D is displacement height (in meters) and $L_{MO}$ is Monin-Obukhov length (in meters). D was set as two third of vegetation height, the latter being derived from in-situ measurements (see Section 2.6.2).
* * *
**R2C35. L 304: Remove "notably".**

Reply to R2C35. Corrected accordingly.

Revisions for R2C35.

The atmospheric stability parameter ξ was most of the time negative, with  few values larger than 0.1, mainly during sunrise or sunset.
* * *
**R2C36. L 314: Please write "land-sea breeze".**

Reply to R2C36. We corrected in order to clarify. We note the study area is located within a Peninsula, and therefore no land-sea breeze are observed.

Revisions for R2C36.

Overall, the analysis of wind direction and micrometeorological conditions indicated that the wind regime did not stem from valley wind or land-sea breeze. Indeed, the wind direction did not depict any diurnal course in relation to anabatic / katabatic flows or to land-sea temperature difference, while the ξ parameter did not correspond to conditions of atmospheric stability with free convection.
* * *
**R2C37. L 329: Please change "was" to "were".**

Reply to R2C37. Corrected accordingly.

Revisions for R2C37.

The vegetation height data indicated that the sonic anemometers and KH20 krypton hygrometers, set up around 2 m above soil surface,  were located above the roughness sublayer.
* * *
**R2C38. L 330: Please rewrite "... the experiment was typified ...".**

Reply to R2C38. Corrected.

Revisions for R2C38.

Indeed, the experimental conditions were typified by neutral or slightly unstable conditions that corresponded to a roughness sublayer extension from the ground up to 1.43 × vegetation height (Pattey et al. 2006).
* * *
**R2C39. L 335/336: The LAI is not necessary in this case and can be removed.**

Reply to R2C39. We disagree. LAI (now GAI in the revised version) was used for splitting the dataset into three periods, see Table 4 and Section 3.2.

No revision for R2C39.
* * *
**R2C40. L 350: Please specify the ratio, e.g. the ratio of the original to filtered time series ...**

Reply to R2C40. Corrected accordingly.

Revisions for R2C40.

The ratio of filtered to original data  ranged between 20 % and 61 %.
* * *
**R2C41. L 358/361: Remove the blank between the number and %.**

Reply to R2C41. Corrected accordingly in the corresponding paragraph and across the manuscript.

Revisions for R2C41.

The ratio of filtered to original data  ranged between 20 % and 61 %.

For  H, the percentages of data belonging to the high-quality classes (I to IV) were 85 %, 84 % and 88 % for fields A, B and C, respectively
* * *
**R2C42. L 436: Remove 'Evaporative fraction' and 'latent heat flux'.**

Reply to R2C42. Please see our reply to R1C3 about acronyms.

Revisions for R2C42. Please see our revisions for R1C3 about acronyms.
* * *
**R2C43. L 450-456: This paragraph should be moved to the Results section.**

Reply to R2C43. We disagree. We deal here with methodological assumptions, not with result analysis. Indeed, energy balance closure is rarely observed.

No revision for R2C43.
* * *
**R2C44. L 587: The expression 'The method performances could be either different or similar before and after splitting ...' is trivial and can be removed.**

Reply to R2C44. We disagree. This shows that splitting is not systemically necessary. This is further discussed in Section 5: " Finally, the performances could be better when splitting the time series on the basis of northwest and south winds, with much lower RMSE values for downslope winds. This was not systematic for the sloping fields, but it was systematic for all methods when applicable, although these methods involved different information for the reconstruction of the missing data. Thus, our study confirmed that it may be  necessary to discriminate upslope and downslope winds when implementing gap-filling methods. "

No revision for R2C44.
* * *
**R2C45. L 637: Better write: 'Overall, the four methods were able to fill all gaps in the time series, ...'.**

Reply to R2C45. Corrected accordingly.

Revisions for R2C45.

Overall, the four methods were able to fill all gaps in  time series, in spite of larger gap occurrences induced by the splitting of the time series on the basis of wind direction.
* * *
**R2C46. L 669: Please write 'The EF method provided lower accuracies'.**

Reply to R2C46. Corrected accordingly.

Revisions for R2C46.

The EF method provided  lower accuracies.
* * *
**R2C47. L 683: Change 'could be' to 'are'.**

Reply to R2C47. We disagree. Please see our reply to R2C44.

No revision for R2C47.
* * *
**R2C48. Figure 4: It would be interesting to provide the same figure (maybe in the supplements) for site B.**

Reply to R2C48. Indeed, this was already done, as indicated in the submitted version, first paragraph of Section 4.3: " We obtained similar results for energy balance closure for field A (Figure 4), field B and C (Figure SP2a and SP2b in supplementary materials)."

No revision for R2C48.
* * *
**R2C49. Table 6: I think the data is more accessible to the reader when presented with a box plot.**

Reply to R2C49. We replaced Table 6 by Figure 5, without indicating RRMSE values which provided very similar patterns than RMSE values. RRMSE values were kept in former Table 6 that was moved within supplementary materials section as Table SP2. Furthermore, New figure 5 relies on barplots rather boxplots, because the latter are meaningless for representing statistical indicators such as RMSE, Bias and $R^2$.

Revisions for R2C49.

- Revisions in paper body

We quantified the retrieval accuracies of the four gap-filling methods by comparing reference data and gap-filling retrievals of  LE over 30-minute intervals for each field and each wind direction (Figure 5 and Table SP2 in supplementary materials).

[ Figure 5 about here.]

- Revisions in figure list and table list.

[Figure]

Figure 5. Accuracy of LE retrievals for the four gap-filling methods (REddyProc labelled as REP, LE - Rn, MLR, EF). Fluxes were calculated over 30-min intervals. Retrieval accuracy is given for each field (A, B, C) and each wind direction (All stands for all data. NW and S stands for northwest and south winds, respectively). Accuracy is quantified using statistical indicators (absolute RMSE, Bias, coefficient of determination $R^2$).

~~Table 6. Accuracy of LE retrievals for the four gap-filling methods (REddyProc, LE - Rn, MLR, EF). Fluxes were calculated over 30-min interval. Retrieval accuracy is given for each field (A, B, C) and each wind direction (NW and S stands for northwest and south winds, respectively) along with the corresponding airflow inclination when applicable (Up and Down stands for upslope and downslope winds, respectively). Accuracy is quantified using statistical indicators (absolute and relative RMSE, Bias, coefficient of determination R2).~~

| - | - | Field A | Field B | Field C | Field A | | Field B | | Field C | |
|---|---|---|---|---|---|---|---|---|---|---|
| - | - | All data | All data | All data | S (Up) | NW (Down) | S (Down) | NW (Up) | S | NW |
| RMSE (W/m²) | REddyProc | 44.8 | 70.5 | 51.9 | 42.3 | 41.4 | 23.3 | 77.2 | 51.1 | 49.1 |
| | LE-Rn | 56.8 | 80.2 | 61.0 | 56.3 | 55.5 | 38.6 | 86.7 | 66.2 | 57.5 |
| | MLR | 58.3 | 61.7 | 59.7 | 55.1 | 55.8 | 37.3 | 61.9 | 61.9 | 57.0 |
| | EF | 57.5 | 87.3 | 62.8 | 48.1 | 56.8 | 42.9 | 98.2 | 63.8 | 57.8 |
| RRMSE (%) | REddyProc | 36 | 57 | 34 | 37 | 32 | 28 | 56 | 42 | 30 |
| | LE-Rn | 46 | 65 | 40 | 50 | 44 | 47 | 63 | 50 | 35 |
| | MLR | 45 | 48 | 37 | 47 | 41 | 45 | 43 | 45 | 34 |
| | EF | 47 | 70 | 41 | 43 | 44 | 52 | 70 | 48 | 36 |
| Bias (W/m²) | REddyProc | 1.34 | 1.13 | 0.65 | 2.14 | 0.90 | 0.96 | 1.58 | 2.20 | 0.80 |
| | LE-Rn | 0.01 | 0.00 | 0.00 | 0.00 | 0.01 | 0.03 | 0.00 | 0.01 | 0.00 |
| | MLR | 0.04 | 0.02 | 0.01 | 0.09 | 0.08 | 0.15 | 0.00 | 0.00 | 0.03 |
| | EF | 16.15 | 6.48 | 15.79 | 10.54 | 19.04 | 0.93 | 8.43 | 12.84 | 17.73 |
| R² | REddyProc | 0.74 | 0.42 | 0.78 | 0.75 | 0.78 | 0.75 | 0.40 | 0.83 | 0.81 |
| | LE-Rn | 0.58 | 0.25 | 0.69 | 0.56 | 0.61 | 0.32 | 0.25 | 0.59 | 0.73 |
| | MLR | 0.58 | 0.35 | 0.72 | 0.59 | 0.62 | 0.36 | 0.38 | 0.65 | 0.75 |
| | EF | 0.69 | 0.29 | 0.74 | 0.75 | 0.71 | 0.52 | 0.24 | 0.83 | 0.80 |

- Revisions in file with supplementary materials.

Table SP2. Accuracy of LE retrievals for the four gap-filling methods (REddyProc, LE - Rn, MLR, EF). Fluxes were calculated over 30-min intervals. Retrieval accuracy is given for each field (A, B, C) and each wind direction (NW and S stands for northwest and south winds, respectively) along with the corresponding airflow inclination when applicable (Up and Down stands for upslope and downslope winds, respectively). Accuracy is quantified using statistical indicators (absolute and relative RMSE, Bias, coefficient of determination $R^2$).

| | | Field A | Field B | Field C | Field A | | Field B | | Field C | |
| | | All data | All data | All data | S (Up) | NW (Down) | S (Down) | NW (Up) | S | NW |
|---|---|---|---|---|---|---|---|---|---|---|
| RMSE (W/m²) | REddyProc | 44.8 | 70.5 | 51.9 | 42.3 | 41.4 | 23.3 | 77.2 | 51.1 | 49.1 |
| | LE-Rn | 56.8 | 80.2 | 61.0 | 56.3 | 55.5 | 38.6 | 86.7 | 66.2 | 57.5 |
| | MLR | 58.3 | 61.7 | 59.7 | 55.1 | 55.8 | 37.3 | 61.9 | 61.9 | 57.0 |
| | EF | 57.5 | 87.3 | 62.8 | 48.1 | 56.8 | 42.9 | 98.2 | 63.8 | 57.8 |
| RRMSE (%) | REddyProc | 36 | 57 | 34 | 37 | 32 | 28 | 56 | 42 | 30 |
| | LE-Rn | 46 | 65 | 40 | 50 | 44 | 47 | 63 | 50 | 35 |
| | MLR | 45 | 48 | 37 | 47 | 41 | 45 | 43 | 45 | 34 |
| | EF | 47 | 70 | 41 | 43 | 44 | 52 | 70 | 48 | 36 |
| Bias (W/m²) | REddyProc | -1.34 | -1.13 | -0.65 | -2.14 | -0.90 | -0.96 | -1.58 | 2.20 | -0.80 |
| | LE-Rn | 0.01 | 0.00 | 0.00 | 0.00 | 0.01 | -0.03 | 0.00 | 0.01 | 0.00 |
| | MLR | 0.04 | -0.02 | 0.01 | -0.09 | 0.08 | -0.15 | 0.00 | 0.00 | 0.03 |
| | EF | -16.15 | -6.48 | -15.79 | -10.54 | -19.04 | -0.93 | -8.43 | -12.84 | -17.73 |
| $R^2$ | REddyProc | 0.74 | 0.42 | 0.78 | 0.75 | 0.78 | 0.75 | 0.40 | 0.83 | 0.81 |
| | LE-Rn | 0.58 | 0.25 | 0.69 | 0.56 | 0.61 | 0.32 | 0.25 | 0.59 | 0.73 |
| | MLR | 0.58 | 0.35 | 0.72 | 0.59 | 0.62 | 0.36 | 0.38 | 0.65 | 0.75 |
| | EF | 0.69 | 0.29 | 0.74 | 0.75 | 0.71 | 0.52 | 0.24 | 0.83 | 0.80 |

---

## Author Comment (AC3) · 10 Apr 2018

Greetings, please find enclosed (supplement) a PDF file that contains our replies to comments from the GI Editorial Review. Best regards.

Please also note the supplement to this comment:
https://www.geosci-instrum-method-data-syst-discuss.net/gi-2017-44/gi-2017-44-AC3-supplement.pdf